# StableDR: Stabilized Doubly Robust Learning for Recommendation on Data Missing Not at Random

**Haoxuan Li**[1]     **Chunyuan Zheng**[2]     **Peng Wu**[3]*
[1]Peking University
[2]University of California, San Diego
[3]Beijing Technology and Business University
hxli@stu.pku.edu.cn,     czheng@ucsd.edu,     pengwu@btbu.edu.cn

## Abstract

In recommender systems, users always choose the favorite items to rate, which leads to data missing not at random and poses a great challenge for unbiased evaluation and learning of prediction models. Currently, the doubly robust (DR) methods have been widely studied and demonstrate superior performance. However, in this paper, we show that DR methods are unstable and have unbounded bias, variance, and generalization bounds to extremely small propensities. Moreover, the fact that DR relies more on extrapolation will lead to suboptimal performance. To address the above limitations while retaining double robustness, we propose a stabilized doubly robust (StableDR) learning approach with a weaker reliance on extrapolation. Theoretical analysis shows that StableDR has bounded bias, variance, and generalization error bound simultaneously under inaccurate imputed errors and arbitrarily small propensities. In addition, we propose a novel learning approach for StableDR that updates the imputation, propensity, and prediction models cyclically, achieving more stable and accurate predictions. Extensive experiments show that our approaches significantly outperform the existing methods.

## 1 Introduction

Modern recommender systems (RSs) are rapidly evolving with the adoption of sophisticated deep learning models (Zhang et al., 2019). However, it is well documented that directly using advanced deep models usually achieves sub-optimal performance due to the existence of various biases in RS (Chen et al., 2020; Wu et al., 2022b), and the biases would be amplified over time (Mansoury et al., 2020; Wen et al., 2022). A large number of debiasing methods have emerged and gradually become a trend. For many practical tasks in RS, such as rating prediction (Schnabel et al., 2016; Wang et al., 2020a; 2019), post-view click-through rate prediction (Guo et al., 2021), post-click conversion rate prediction (Zhang et al., 2020; Dai et al., 2022), and uplift modeling (Saito et al., 2019; Sato et al., 2019; 2020), a critical challenge is to combat the selection bias and confounding bias that leading to significantly difference between the trained sample and the targeted population (Hernán & Robins, 2020). Various methods were designed to address this problem and among them, doubly robust (DR) methods (Wang et al., 2019; Zhang et al., 2020; Chen et al., 2021; Dai et al., 2022; Ding et al., 2022) play the dominant role due to their better performance and theoretical properties.

The success of DR is attributed to its double robustness and joint-learning technique. However, the DR methods still have many limitations. Theoretical analysis shows that inverse probability scoring (IPS) and DR methods may have infinite bias, variance, and generalization error bounds, in the presence of extremely small propensity scores (Schnabel et al., 2016; Wang et al., 2019; Guo et al., 2021; Li et al., 2023b). In addition, due to the fact that users are more inclined to evaluate the preferred items, the problem of data missing not at random (MNAR) often occurs in RS. This would cause selection bias and results in inaccuracy for methods that more rely on extrapolation, such as error imputation based (EIB) (Marlin et al., 2007; Steck, 2013) and DR methods.

---

*Corresponding author.

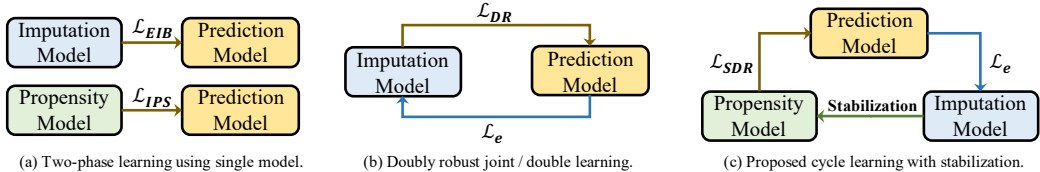

(a) Two-phase learning using single model.    (b) Doubly robust joint / double learning.    (c) Proposed cycle learning with stabilization.

Figure 1: During the training of updating a prediction model, two-phase learning (Marlin et al., 2007; Steck, 2013; Schnabel et al., 2016) uses a fixed imputation/propensity model (Left), whereas DR-JL (Wang et al., 2019), MRDR-DL (Guo et al., 2021), and AutoDebias (Chen et al., 2021) uses alternative learning between the imputation/propensity and the prediction model (Middle). The proposed learning approach updates the three models cyclically with stabilization (Right).

To overcome the above limitations while maintaining double robustness, we propose a stabilized doubly robust (SDR) estimator with a weaker reliance on extrapolation, which reduces the negative impact of extrapolation and MNAR effect on the imputation model. Through theoretical analysis, we demonstrate that the SDR has bounded bias and generalization error bound for arbitrarily small propensities, which further indicates that the SDR can achieve more stable predictions.

Furthermore, we propose a novel cycle learning approach for SDR. Figure 1 shows the differences between the proposed cycle learning of SDR and the existing unbiased learning approaches. Two-phase learning (Marlin et al., 2007; Steck, 2013; Schnabel et al., 2016) first obtains an imputation/propensity model to estimate the ideal loss and then updates the prediction model by minimizing the estimated loss. DR-JL (Wang et al., 2019), MRDR-DL (Guo et al., 2021), and AutoDebias (Chen et al., 2021) alternatively update the model used to estimate the ideal loss and the prediction model. The proposed learning method cyclically uses different losses to update the three models with the aim of achieving more stable and accurate prediction results. We have conducted extensive experiments on two real-world datasets, and the results show that the proposed approach significantly improves debiasing and convergence performance compared to the existing methods.

## 2 PRELIMINARIES

### 2.1 PROBLEM SETTING

In RS, due to the fact that users are more inclined to evaluate the preferred items, the collected ratings are always missing not at random (MNAR). We formulate the data MNAR problem using the widely adopted potential outcome framework (Neyman, 1990; Imbens & Rubin, 2015). Let $\mathcal{U} = \{1, 2, ..., U\}$, $\mathcal{I} = \{1, 2, ..., I\}$ and $\mathcal{D} = \mathcal{U} \times \mathcal{I}$ be the index sets of users, items, all user-item pairs. For each $(u, i) \in \mathcal{D}$, we have a treatment $o_{u,i} \in \{0, 1\}$, a feature vector $x_{u,i}$, and an observed rating $r_{u,i}$, where $o_{u,i} = 1$ if user $u$ rated the item $i$ in the logging data, $o_{u,i} = 0$ if the rating is missing. Let $r_{u,i}(1)$ is defined as the be the rating that would be observed if item $i$ had been rated by user $u$, which is observable only for $\mathcal{O} = \{(u, i) \mid (u, i) \in \mathcal{D}, o_{u,i} = 1\}$. Many tasks in RS can be formulated by predicting the potential outcome $r_{u,i}(1)$ using feature $x_{u,i}$ for each $(u, i)$.

Let $\hat{r}_{u,i}(1) = f(x_{u,i}; \phi)$ be a prediction model with parameters $\phi$. If all the potential outcomes $\{r_{u,i}(1) : (u, i) \in \mathcal{D}\}$ were observed, the ideal loss function for solving parameters $\phi$ is given as

$$\mathcal{L}_{ideal}(\phi) = |\mathcal{D}|^{-1} \sum_{(u,i) \in \mathcal{D}} e_{u,i},$$

where $e_{u,i}$ is the prediction error, such as the squared loss $e_{u,i} = (\hat{r}_{u,i}(1) - r_{u,i}(1))^2$. $\mathcal{L}_{ideal}(\phi)$ can be regarded as a benchmark of unbiased loss function, even though it is infeasible due to the missingness of $\{r_{u,i}(1) : o_{u,i} = 0\}$. As such, a variety of methods are developed through approximating $\mathcal{L}_{ideal}(\phi)$ to address the selection bias, in which the propensity-based estimators show the relatively superior performance (Schnabel et al., 2016; Wang et al., 2019), and the IPS and DR estimators are

$$\mathcal{E}_{IPS} = |\mathcal{D}|^{-1} \sum_{(u,i) \in \mathcal{D}} \frac{o_{u,i} e_{u,i}}{\hat{p}_{u,i}} \quad \text{and} \quad \mathcal{E}_{DR} = |\mathcal{D}|^{-1} \sum_{(u,i) \in \mathcal{D}} \left[ \hat{e}_{u,i} + \frac{o_{u,i}(e_{u,i} - \hat{e}_{u,i})}{\hat{p}_{u,i}} \right],$$

where $\hat{p}_{u,i}$ is an estimate of propensity score $p_{u,i} := \mathbb{P}(o_{u,i} = 1 | x_{u,i})$, $\hat{e}_{u,i}$ is an estimate of $e_{u,i}$.

## 2.2 RELATED WORK

**Debiased learning in recommendation.** The data collected in RS suffers from various biases (Chen et al., 2020; Wu et al., 2022b), which are entangled with the true preferences of users and pose a great challenge to unbiased learning. There is increasing interest in coping with different biases in recent years (Zhang et al., 2021; Ai et al., 2018; Liu et al., 2016; Liu et al., 2021). Schnabel et al. (2016) proposed using inverse propensity score (IPS) and self-normalized IPS (SNIPS) methods to address the selection bias on data missing not at random, Saito (2019) and Saito et al. (2020) extended them to implicit feedback data. Marlin et al. (2007) and Steck (2013) derived an error imputation-based (EIB) unbiased learning method. These three approaches adopt two-phase learning (Wang et al., 2021), which first learns a propensity/imputation model and then applies it to construct an unbiased estimator of the ideal loss to train the recommendation model. A doubly robust joint learning (DR-JL) method (Wang et al., 2019) was proposed by combining the IPS and EIB approaches. Subsequently, strands of enhanced joint learning methods were developed, including MRDR (Guo et al., 2021), Multi-task DR (Zhang et al., 2020), DR-MSE (Dai et al., 2022), BRD-DR (Ding et al., 2022), TDR Li et al. (2023b), uniform data-aware methods (Bonner & Vasile, 2018; Liu et al., 2020; Chen et al., 2021; Wang et al., 2021; Li et al., 2023c) that aimed to seek better recommendation strategies by leveraging a small uniform dataset, and multiple robust method (Li et al., 2023a) that specifies multiple propensity and imputation models and achieves unbiased learning if any of the propensity models, imputation models, or even a linear combination of these models can accurately estimate the true propensities or prediction errors. Chen et al. (2020) reviewed various biases in RS and discussed the recent progress on debiasing tasks. Wu et al. (2022b) established the connections between the biases in causal inference and the biases, thereby presenting the formal causal definitions for RS.

**Stabilized causal effect estimation.** The proposed method builds on the stabilized average causal effect estimation approaches in causal inference. Molenberghs et al. (2015) summarized the limitations of doubly robust methods, including unstable to small propensities (Kang & Schafer, 2007; Wu et al., 2022a), unboundedness (van der Laan & Rose, 2011), and large variance (Tan, 2007). These issues inspired a series of stabilized causal effect estimation methods in statistics (Kang & Schafer, 2007; Bang & Robins, 2005; van der Laan & Rose, 2011; Molenberghs et al., 2015). Unlike previous works that focused only on achieving learning with unbiasedness in RS, this paper provides a new perspective to develop doubly robust estimators with much more stable statistical properties.

## 3 STABILIZED DOUBLY ROBUST ESTIMATOR

In this section, we elaborate the limitations of DR methods and propose a stabilized DR (SDR) estimator with a weaker reliance on extrapolation. Theoretical analysis shows that SDR has bounded bias and generalization error bound for arbitrarily small propensities, while IPS and DR don't.

## 3.1 MOTIVATION

Even though DR estimator has double robustness property, its performance could be significantly improved if the following three stabilization aspects can be enhanced.

**More stable to small propensities.** As shown in Schnabel et al. (2016), Wang et al. (2019) and Guo et al. (2021), if there exist some extremely small estimated propensity scores, the IPS/DR estimator and its bias, variance, and tail bound are unbounded, deteriorating the prediction accuracy. What's more, such problems are widespread in practice, given the fact that there are many long-tailed users and items in RS, resulting in the presence of extreme propensities.

**More stable through weakening extrapolation.** DR relies more on extrapolation because the imputation model in DR is learned from the exposed events $\mathcal{O}$ and extrapolated to the unexposed events. If the distributional disparity of $e_{u,i}$ on $o_{u,i} = 0$ and $o_{u,i} = 1$ is large, the imputed errors are likely to be inaccurate on the unexposed events and incur bias of DR. Therefore, it is beneficial to reduce bias if we can develop an enhanced DR method with weaker reliance on extrapolation.

**More stable training process of updating a prediction model.** In general, alternating training between models results in better performance. From Figure 1, Wang et al. (2019) proposes joint learning for DR, alternatively updating the error imputation and prediction models with given estimated propensities. Double learning (Guo et al., 2021) further incorporates parameter sharing between the

imputation and prediction models. Bi-level optimization (Wang et al., 2021; Chen et al., 2021) can be viewed as alternately updating the prediction model and the other parameters used to estimate the loss. To the best of our knowledge, this is the first paper that proposes a algorithm to update the three models (i.e., error imputation model, propensity model, and prediction model) separately using different optimizers, which may resulting in more stable and accurate rating predictions.

## 3.2 STABILIZED DOUBLY ROBUST ESTIMATOR

We propose a stabilized doubly robust (SDR) estimator that has a weaker dependence on extrapolation and is robust to small propensities. The SDR estimator consists of the following three steps.

**Step 1 (Initialize imputed errors).** Pre-train imputation model $\hat{e}_{u,i}$, let $\hat{\mathcal{E}} \triangleq |\mathcal{D}|^{-1} \sum_{(u,i)\in\mathcal{D}} \hat{e}_{u,i}$.

**Step 2 (Learn constrained propensities).** Learn a propensity model $\hat{p}_{u,i}$ satisfying

$$\frac{1}{|\mathcal{D}|} \sum_{(u,i)\in\mathcal{D}} \frac{o_{u,i}}{\hat{p}_{u,i}} \left( \hat{e}_{u,i} - \hat{\mathcal{E}} \right) = 0. \tag{1}$$

**Step 3 (SDR estimator).** The SDR estimator is given as

$$\mathcal{E}_{SDR} = \sum_{(u,i)\in\mathcal{D}} \frac{o_{u,i}e_{u,i}}{\hat{p}_{u,i}} \bigg/ \sum_{(u,i)\in\mathcal{D}} \frac{o_{u,i}}{\hat{p}_{u,i}} \triangleq \sum_{(u,i)\in\mathcal{D}} w_{u,i}e_{u,i},$$

where $w_{u,i} = \frac{o_{u,i}}{\hat{p}_{u,i}} \big/ \sum_{(u,i)\in\mathcal{D}} \frac{o_{u,i}}{\hat{p}_{u,i}}$. It can be seen that SDR estimator has the same form as SNIPS estimator, but the propensities are learned differently. In SDR, the estimation of propensity model relies on the imputed errors, whereas not in SNIPS.

Each step in the construction of SDR estimator plays a different role. Specifically, the Step 2 is designed to enable double robustness property as shown in Theorem 1 (see Appendix A.1 for proofs).

**Theorem 1 (Double Robustness).** *$\mathcal{E}_{SDR}$ is an asymptotically unbiased[1] estimator of $\mathcal{L}_{ideal}$, when either the learned propensities $\hat{p}_{u,i}$ or the imputed errors $\hat{e}_{u,i}$ are accurate for all user-item pairs.*

We provide an intuitive way to illustrate the rationale of SDR. **On the one hand**, if the propensities can be accurately estimated (i.e., $\hat{p}_{u,i} = p_{u,i}$) by using a common model (e.g., logistic regression) without imposing constraint (1). Then the expectation of the left hand side of constraint (1) becomes

$$\mathbb{E}_{\mathcal{O}}\Big[ \frac{1}{|\mathcal{D}|} \sum_{(u,i)\in\mathcal{D}} \frac{o_{u,i}}{\hat{p}_{u,i}} \left( \hat{e}_{u,i} - \hat{\mathcal{E}} \right) \Big] = \frac{1}{|\mathcal{D}|} \sum_{(u,i)\in\mathcal{D}} \left( \hat{e}_{u,i} - \hat{\mathcal{E}} \right) \equiv 0,$$

which indicates the constraint (1) always holds as the sample size goes to infinity by the strong law of large numbers[2], irrespective of the accuracy of the imputed errors $\hat{e}_{u,i}$. This implies that the constraint (1) *imposes almost no restriction* on the estimation of propensities. In this case, the SDR estimator is almost equivalent to the original SNIPS estimator. **On the other hand**, if the propensities *cannot* be accurately estimated by using a common model, but the imputed errors are accurate (i.e., $\hat{e}_{u,i} = e_{u,i}$). In this case, $\hat{\mathcal{E}}$ is an unbiased estimator. Specifically, $\mathcal{E}_{SDR}$ satisfies

$$\frac{1}{|\mathcal{D}|} \sum_{(u,i)\in\mathcal{D}} \frac{o_{u,i}(e_{u,i} - \mathcal{E}_{SDR})}{\hat{p}_{u,i}} = 0. \tag{2}$$

Combining the constraint (1) and equation (2) gives

$$\frac{1}{|\mathcal{D}|} \sum_{(u,i)\in\mathcal{D}} \left[ \frac{o_{u,i}(e_{u,i} - \hat{e}_{u,i})}{\hat{p}_{u,i}} + \frac{o_{u,i}(\hat{\mathcal{E}} - \mathcal{E}_{SDR})}{\hat{p}_{u,i}} \right] = 0, \tag{3}$$

where the first term equals to 0 if $\hat{e}_{u,i} = e_{u,i}$, it implies that $\mathcal{E}_{SDR} = \hat{\mathcal{E}}$, then the unbiasedness of $\mathcal{E}_{SDR}$ follows immediately from the unbiasedness of $\hat{\mathcal{E}}$.

---

[1]Asymptotically unbiased means unbiasedness as the sample size goes to infinity.
[2]This is the reason why we adopt the notation of "asymptotically unbiased".

In addition, Step 3 is designed for two main reasons to achieve stability. First, $\mathcal{E}_{SDR}$ is more robust to extrapolation compared with DR. This is because the propensities are learned from the entire data and thus have less requirement on extrapolation. Second, $\mathcal{E}_{SDR}$ is more stable to small propensities, since the self-normalization imposes the weight $w_{u,i}$ to fall on the interval [0,1].

In summary, forcing the propensities to satisfy the constraint (1) makes the SDR estimator not only doubly robust, but also captures the advantages of both SNIPS and DR estimators. The design of SDR enables the constrained propensities to adaptively find the direction of debiasing if either the learned propensities without imposing constraint (1) or the imputed errors are accurate.

### 3.3 THEORETICAL ANALYSIS OF STABLENESS

Through theoretical analysis, we note that previous debiasing estimators such as IPS (Schnabel et al., 2016) and DR-based methods (Wang et al., 2019; Guo et al., 2021) tend to have infinite biases, variances, tail bound, and corresponding generalization error bounds, in the presence of extremely small estimated propensities. Remarkably, the proposed SDR estimator doesn't suffer from such problems and is stable to arbitrarily small propensities, as shown in the following Theorems (see Appendixes A.2, A.3 and A.4 for proofs).

**Theorem 2** (Bias of SDR). *Given imputed errors $\hat{e}_{u,i}$ and learned propensities $\hat{p}_{u,i}$ satisfying the stabilization constraint (1), with $\hat{p}_{u,i} > 0$ for all user-item pairs, the bias of $\mathcal{E}_{SDR}$ is*

$$\text{Bias}(\mathcal{E}_{SDR}) = \left| \frac{1}{|\mathcal{D}|} \sum_{(u,i)\in\mathcal{D}} \left( \delta_{u,i} - \frac{\sum_{(u,i)\in\mathcal{D}} \delta_{u,i} p_{u,i}/\hat{p}_{u,i}}{\sum_{(u,i)\in\mathcal{D}} p_{u,i}/\hat{p}_{u,i}} \right) \right| + O(|\mathcal{D}|^{-1}),$$

*where $\delta_{u,i} = e_{u,i} - \hat{e}_{u,i}$ is the error deviation.*

Theorem 2 shows the bias of the SDR estimator consisting of a dominant term given by the difference between $\delta_{u,i}$ and its weighted average, and a negligible term of order $O(|\mathcal{D}|^{-1})$. The fact that the $\delta_{u,i}$ and its convex combinations are bounded, shows that the bias is bounded for arbitrarily small $\hat{p}_{u,i}$. Compared to the $\text{Bias}(\mathcal{E}_{IPS}) = |\mathcal{D}|^{-1} |\sum_{u,i\in\mathcal{D}} (\hat{p}_{u,i} - p_{u,i})e_{u,i}/\hat{p}_{u,i}|$ and $\text{Bias}(\mathcal{E}_{DR}) = |\mathcal{D}|^{-1} |\sum_{u,i\in\mathcal{D}} (\hat{p}_{u,i} - p_{u,i})\delta_{u,i}/\hat{p}_{u,i}|$, it indicates that IPS and DR will have extremely large bias when there exists an extremely small $\hat{p}_{u,i}$.

**Theorem 3** (Variance of SDR). *Under the conditions of Theorem 2, the variance of $\mathcal{E}_{SDR}$ is*

$$\text{Var}(\mathcal{E}_{SDR}) = \frac{\sum_{(u,i)\in\mathcal{D}} p_{u,i}(1-p_{u,i})h_{u,i}^2/\hat{p}_{u,i}^2}{\left( \sum_{(u,i)\in\mathcal{D}} p_{u,i}/\hat{p}_{u,i} \right)^2} + O(|\mathcal{D}|^{-2}),$$

*where $h_{u,i} = (e_{u,i} - \hat{e}_{u,i}) - \sum_{(u,i)\in\mathcal{D}}\{p_{u,i}(e_{u,i} - \hat{e}_{u,i})/\hat{p}_{u,i}\}/\sum_{(u,i)\in\mathcal{D}}\{p_{u,i}/\hat{p}_{u,i}\}$ is a bounded difference between $e_{u,i} - \hat{e}_{u,i}$ and its weighted average.*

Theorem 3 shows the variance of the SDR estimator consisting of a dominant term and a negligible term of order $O(|\mathcal{D}|^{-2})$. The boundedness of the variance for arbitrarily small $\hat{p}_{u,i}$ is given directly from the fact that SDR has a bounded range given by the self-normalized form. Compared to the $\text{Var}(\mathcal{E}_{IPS}) = |\mathcal{D}|^{-2} \sum_{u,i\in\mathcal{D}} p_{u,i}(1-p_{u,i})e_{u,i}^2/\hat{p}_{u,i}^2$ and $\text{Var}(\mathcal{E}_{DR}) = |\mathcal{D}|^{-2} \sum_{u,i\in\mathcal{D}} p_{u,i}(1-p_{u,i})(e_{u,i} - \hat{e}_{u,i})^2/\hat{p}_{u,i}^2$, it indicates that IPS and DR will have extremely large variance (tend to infinity) when there exist an extremely small $\hat{p}_{u,i}$ (tends to 0).

**Theorem 4** (Tail Bound of SDR). *Under the conditions of Theorem 2, for any prediction model, with probability $1 - \eta$, the deviation of $\mathcal{E}_{SDR}$ from its expectation has the following tail bound*

$$|\mathcal{E}_{SDR} - \mathbb{E}_{\mathcal{O}}(\mathcal{E}_{SDR})| \leq \sqrt{\frac{1}{2}\log\left(\frac{4}{\eta}\right) \sum_{(u,i)\in\mathcal{D}} \frac{(\delta_{\max} - \delta_{u,i})^2 + (\delta_{u,i} - \delta_{\min})^2}{\{1 + \hat{p}_{u,i}(\sum_{\mathcal{D}\backslash(u,i)} p_{u,i}/\hat{p}_{u,i} - \epsilon')\}^2}}$$

*where $\delta_{\min} = \min_{(u,i)\in\mathcal{D}} \delta_{u,i}$, $\delta_{\max} = \max_{(u,i)\in\mathcal{D}} \delta_{u,i}$, $\epsilon' = \sqrt{\log(4/\eta)/2 \cdot \sum_{\mathcal{D}\backslash(u,i)} 1/\hat{p}_{u,i}^2}$, and $\mathcal{D} \backslash (u,i)$ is the set of $\mathcal{D}$ excluding the element $(u,i)$.*

Note that $\sum_{\mathcal{D}\setminus(u,i)} p_{u,i}/\hat{p}_{u,i} = O(|\mathcal{D}|)$ and $\epsilon' = O(|\mathcal{D}|^{1/2})$ in Theorem 4, it follows that the tail bound of the SDR estimator converges to 0 for large samples. In addition, the tail bound is bounded for arbitrarily small $\hat{p}_{u,i}$. Compared to the tail bound of IPS and DR, with probability $1-\eta$, we have

$$|\mathcal{E}_{IPS} - \mathbb{E}_{\mathcal{O}}[\mathcal{E}_{IPS}]| \leq \sqrt{\frac{\log(2/\eta)}{2|\mathcal{D}|^2} \sum_{(u,i)\in\mathcal{D}} \left(\frac{e_{u,i}}{\hat{p}_{u,i}}\right)^2}, |\mathcal{E}_{DR} - \mathbb{E}_{\mathcal{O}}[\mathcal{E}_{DR}]| \leq \sqrt{\frac{\log(2/\eta)}{2|\mathcal{D}|^2} \sum_{(u,i)\in\mathcal{D}} \left(\frac{\delta_{u,i}}{\hat{p}_{u,i}}\right)^2},$$

which are both unbounded when $\hat{p}_{u,i} \to 0$. For SDR in the prediction model training phase, the boundedness of the generalization error bound (see Theorem 5 in Appendix A.5) follows immediately from the boundedness of the bias and tail bound. The above analysis demonstrates that SDR can comprehensively mitigate the negative effects caused by extreme propensities and results in more stable predictions. Theorems 2-5 are stated under the constraint (1). If we estimate the propensities with constraint (1), but finally constraint (1) somehow doesn't hold exactly, the associated bias, variance, and generalization error bound of SDR are presented in Appendix B.

## 4 CYCLE LEARNING WITH STABILIZATION

In this section, we propose a novel SDR-based cycle learning approach, that not only exploits the stable statistical properties of the SDR estimator itself, but also carefully designs the updating process among various models to achieve higher stability. In general, inspired by the idea of value iteration in reinforcement learning (Sutton & Barto, 2018), alternatively updating the model tends to achieve better predictive performance, as existing debiasing training approaches suggested (Wang et al., 2019; Guo et al., 2021; Chen et al., 2021). As shown in Figure 1, the proposed approach dynamically interacts with three models, utilizing the propensity model and imputation model simultaneously in a differentiated way, which can be regarded as an extension of these methods. In cycle learning, given pre-trained propensities, the inverse propensity weighted imputation error loss is used to first obtain an imputation model, and then take the constraint (1) as the regularization term to train a stabilized propensity model and ensure the double robustness of SDR. Finally, the prediction model is updated by minimizing the SDR loss and used to readjust the imputed errors. By repeating the above update processes cyclically, the cycle learning approach can fully utilize and combine the advantages of the three models to achieve more accurate rating predictions.

Specifically, the data MNAR leads to the presence of missing $r_{u,i}(1)$, so that all $e_{u,i}$ cannot be used directly. Therefore, we obtain imputed errors by learning a pseudo-labeling model $\tilde{r}_{u,i}(1)$ parameterized by $\beta$, and the imputed errors $\hat{e}_{u,i} = \text{CE}(\tilde{r}_{u,i}(1), \hat{r}_{u,i}(1))$ are updated by minimizing

$$\mathcal{L}_e(\phi,\alpha,\beta) = |\mathcal{D}|^{-1} \sum_{(u,i)\in\mathcal{D}} \frac{o_{u,i}(\hat{e}_{u,i} - e_{u,i})^2}{\pi(x_{u,i};\alpha)} + \lambda_e\|\beta\|_F^2,$$

where $e_{u,i} = \text{CE}(r_{u,i}(1), \hat{r}_{u,i}(1))$, $\lambda_e \geq 0$, $\hat{p}_{u,i} = \pi(x_{u,i};\alpha)$ is the propensity model, $\|\cdot\|_F^2$ is the Frobenius norm. For each observed ratings, the inverse of the estimated propensities are used for weighting to account for MNAR effects. Next, we consider two methods for estimating propensity scores, which are Naive Bayes with Laplace smoothing and logistic regression. The former provides a wide range of opportunities for achieving stability constraint (1) through the selection of smoothing coefficients. The latter requires user and item embeddings, which are obtained by employing MF before performing cycle learning. The learned propensities need to both satisfy the accuracy, which is evaluated with cross entropy, and meet the constraint (1) for stabilization and double robustness. The propensity model $\pi(x_{u,i};\alpha)$ is updated by using the loss $\mathcal{L}_{ce}(\phi,\alpha,\beta) + \eta \cdot \mathcal{L}_{stable}(\phi,\alpha,\beta)$, where $\mathcal{L}_{ce}(\phi,\alpha,\beta)$ is cross entropy loss of propensity model and

$$\mathcal{L}_{stable}(\phi,\alpha,\beta) = |\mathcal{D}|^{-1}\bigg\{ \sum_{(u,i)\in\mathcal{D}} \frac{o_{u,i}}{\pi(x_{u,i};\alpha)}\Big(\hat{e}_{u,i} - \hat{\mathcal{E}}\Big)\bigg\}^2 + \lambda_{stable}\|\alpha\|_F^2,$$

where $\lambda_{stable} \geq 0$, and $\eta$ is a hyper-parameter for trade-off. Finally, the prediction model $f(x_{u,i};\phi)$ is updated by minimizing the SDR loss

$$\mathcal{L}_{sdr}(\phi,\alpha,\beta) = \Big[ \sum_{(u,i)\in\mathcal{D}} \frac{o_{u,i}e_{u,i}}{\pi(x_{u,i};\alpha)}\Big]\Big/\Big[ \sum_{(u,i)\in\mathcal{D}} \frac{o_{u,i}}{\pi(x_{u,i};\alpha)}\Big] + \lambda_{sdr}\|\phi\|_F^2,$$

---

**Algorithm 1:** The Proposed Stable DR (MRDR) Cycle Learning, Stable-DR (MRDR)

---

**Input:** observed ratings $\mathbf{Y}^o$, and $\eta, \lambda_e, \lambda_{stable}, \lambda_{sdr} \geq 0$

1 **while** *stopping criteria is not satisfied* **do**
2     **for** *number of steps for training the imputation model* **do**
3         Sample a batch of user-item pairs $\{(u_j, i_j)\}_{j=1}^J$ from $\mathcal{O}$;
4         Update $\beta$ by descending along the gradient $\nabla_\beta \mathcal{L}_e(\phi, \alpha, \beta)$;
5     **end**
6     **for** *number of steps for training the propensity model* **do**
7         Sample a batch of user-item pairs $\{(u_k, i_k)\}_{k=1}^K$ from $\mathcal{D}$;
8         Calculate the gradient of propensity cross entropy error $\nabla_\alpha \mathcal{L}_{ce}(\phi, \alpha, \beta)$;
9         Calculate the gradient of propensity stable constraint (1) $\nabla_\alpha \mathcal{L}_{stable}(\phi, \alpha, \beta)$;
10         Update $\alpha$ by descending along the gradient $\nabla_\alpha \mathcal{L}_{ce}(\phi, \alpha, \beta) + \eta \cdot \nabla_\alpha \mathcal{L}_{stable}(\phi, \alpha, \beta)$
11     **end**
12     **for** *number of steps for training the prediction model* **do**
13         Sample a batch of user-item pairs $\{(u_l, i_l)\}_{l=1}^L$ from $\mathcal{O}$;
14         Update $\phi$ by descending along the gradient $\nabla_\phi \mathcal{L}_{sdr}(\phi, \alpha, \beta)$;
15     **end**
16 **end**

---

where the first term is equivalent to the left hand side of equation (3), and $\lambda_{sdr} \geq 0$. In cycle learning, the updated prediction model will be used for re-update the imputation model using the next sample batch. Notably, the designed algorithm strictly follows the proposed SDR estimator in Section 3.2. From Figure 1 and Alg. 1, our algorithm first updates imputed errors $\hat{e}$ by Step 1, and then learns a propensity $\hat{p}$ based on learned $\hat{e}$ to satisfy the constraint (1) in Step 2. The main purpose of the first two steps is to ensure that the SDR estimator in Step 3 has double robustness and has a lower extrapolation dependence compared to the previous DR methods. Finally, from Step 3 we update the predicted rating $\hat{r}$ by minimizing the estimation of the ideal loss using the proposed SDR estimator. For the next round, instead of re-initializing, Step 1 updates the imputed errors $\hat{e}$ according to the new prediction model, then Step 2 re-updates the constrained propensities $\hat{p}$, and then uses Step 3 to update the prediction model $\hat{r}$ again, and so on. We summarized the cycle learning approach in Alg. 1.

## 5 REAL-WORLD EXPERIMENTS

In this section, several experiments are conducted to evaluate the proposed methods on two real-world benchmark datasets. We conduct experiments to answer the following questions:

**RQ1.** Do the proposed Stable-DR and Stable-MRDR approaches improve in debiasing performance compared to the existing studies?

**RQ2.** Do our methods stably perform well under the various propensity models?

**RQ3.** How does the performance of our method change under different strengths of the stabilization constraint?

### 5.1 EXPERIMENTAL SETUP

**Dataset and preprocessing.** To answer the above RQs, we need to use the datasets that contain both MNAR ratings and missing-at-random (MAR) ratings. Following the previous studies (Schnabel et al., 2016; Wang et al., 2019; Guo et al., 2021; Chen et al., 2021), we conduct experiments on the two commonly used datasets: **Coat**[3] contains ratings from 290 users to 300 items. Each user evaluates 24 items, containing 6,960 MNAR ratings in total. Meanwhile, each user evaluates 16 items randomly, which generates 4,640 MAR ratings. **Yahoo! R3**[4] contains totally 311,704 MNAR and 54,000 MAR ratings from 15,400 users to 1,000 items. **Baselines.** In our experiments, we take **Matrix Factorization (MF)** (Koren et al., 2009), **Sparse LInear Method (SLIM)** (Ning & Karypis,

---

[3]https://www.cs.cornell.edu/~schnabts/mnar/
[4]http://webscope.sandbox.yahoo.com/

Table 1: Performance on Coat and Yahoo!R3, using MF, SLIM, and NCF as the base models.

| Dataset | Coat | | | | Yahoo!R3 | | | |
|---|---|---|---|---|---|---|---|---|
| Method | MSE | AUC | N@5 | N@10 | MSE | AUC | N@5 | N@10 |
| MF | 0.2428 | 0.7063 | 0.6025 | 0.6774 | 0.2500 | 0.6722 | 0.6374 | 0.7634 |
| + IPS | 0.2316 | 0.7166 | 0.6184 | 0.6897 | 0.2194 | 0.6742 | 0.6304 | 0.7556 |
| + SNIPS | 0.2333 | 0.7070 | 0.6222 | 0.6851 | **0.1931** | 0.6831 | 0.6348 | 0.7608 |
| + AS-IPS | **0.2121** | 0.7180 | 0.6160 | 0.6824 | 0.2391 | 0.6770 | 0.6364 | 0.7601 |
| + CVIB | 0.2195 | 0.7239 | 0.6285 | 0.6947 | 0.2625 | 0.6853 | 0.6513 | 0.7729 |
| + DR | 0.2298 | 0.7132 | 0.6243 | 0.6918 | 0.2093 | 0.6873 | 0.6574 | 0.7741 |
| + DR-JL | 0.2254 | 0.7209 | 0.6252 | 0.6961 | 0.2194 | 0.6863 | 0.6525 | 0.7701 |
| + **Stable-DR (Ours)** | 0.2159 | **0.7508** | **0.6511** | **0.7073** | 0.2090 | **0.6946** | **0.6620** | **0.7786** |
| + MRDR-JL | 0.2252 | 0.7318 | 0.6375 | 0.6989 | 0.2173 | 0.6830 | 0.6437 | 0.7652 |
| + **Stable-MRDR (Ours)** | **0.2076** | **0.7548** | **0.6532** | **0.7105** | **0.2081** | **0.6915** | **0.6585** | **0.7757** |
| SLIM | 0.2419 | 0.7074 | 0.7064 | 0.7650 | 0.2126 | 0.6636 | 0.7190 | 0.8134 |
| + IPS | 0.2411 | 0.7058 | 0.7235 | 0.7644 | 0.2046 | 0.6583 | 0.7285 | 0.8244 |
| + SNIPS | 0.2420 | 0.7071 | 0.7369 | 0.7672 | 0.2155 | 0.6720 | 0.7303 | 0.8227 |
| + AS-IPS | **0.2133** | 0.7105 | 0.6238 | 0.6975 | **0.1946** | 0.6769 | 0.6508 | 0.7702 |
| + CVIB | 0.2413 | 0.7108 | 0.7214 | 0.7638 | **0.2024** | 0.6790 | 0.7335 | 0.8221 |
| + DR | **0.2334** | 0.7064 | 0.7267 | 0.7649 | 0.2054 | 0.6771 | 0.7344 | 0.8248 |
| + DR-JL | 0.2407 | 0.7090 | 0.7279 | 0.7655 | 0.2044 | 0.6792 | 0.7360 | 0.8260 |
| + **Stable-DR (Ours)** | 0.2356 | **0.7201** | **0.7389** | **0.7724** | 0.2080 | **0.6874** | **0.7473** | **0.8349** |
| + MRDR-JL | 0.2409 | 0.7074 | 0.7329 | 0.7679 | 0.2016 | 0.6791 | 0.7338 | 0.8239 |
| + **Stable-MRDR (Ours)** | 0.2369 | **0.7148** | **0.7378** | **0.7711** | 0.2086 | **0.6842** | **0.7435** | **0.8308** |
| NCF | 0.2050 | 0.7670 | 0.6228 | 0.6954 | 0.3215 | 0.6782 | 0.6501 | 0.7672 |
| + IPS | 0.2042 | 0.7646 | 0.6327 | 0.7054 | 0.1777 | 0.6719 | 0.6548 | 0.7703 |
| + SNIPS | 0.1904 | 0.7707 | 0.6271 | 0.7062 | 0.1694 | 0.6903 | 0.6693 | 0.7807 |
| + AS-IPS | 0.2061 | 0.7630 | 0.6156 | 0.6983 | 0.1715 | 0.6879 | 0.6620 | 0.7769 |
| + CVIB | 0.2042 | 0.7655 | 0.6176 | 0.6946 | 0.3088 | 0.6715 | 0.6669 | 0.7793 |
| + DR | 0.2081 | 0.7578 | 0.6119 | 0.6900 | 0.1705 | 0.6886 | 0.6628 | 0.7768 |
| + DR-JL | 0.2115 | 0.7600 | 0.6272 | 0.6967 | 0.2452 | 0.6818 | 0.6516 | 0.7678 |
| + **Stable-DR (Ours)** | **0.1896** | **0.7712** | **0.6337** | **0.7095** | **0.1664** | **0.6907** | **0.6756** | **0.7861** |
| + MRDR-JL | 0.2046 | 0.7609 | 0.6182 | 0.6992 | 0.2367 | 0.6778 | 0.6465 | 0.7664 |
| + **Stable-MRDR (Ours)** | **0.1899** | **0.7710** | **0.6380** | **0.7082** | **0.1671** | **0.6910** | **0.6734** | **0.7846** |

2011), and **Neural Collaborative Filtering (NCF)** (He et al., 2017) as the base model respectively, and compare against the proposed methods with the following baselines: **Base Model**, **IPS** (Saito et al., 2020; Schnabel et al., 2016), **SNIPS** (Swaminathan & Joachims, 2015), **IPS-AT** (Saito, 2020), **CVIB** (Wang et al., 2020b), **DR** (Saito, 2020), **DR-JL** (Wang et al., 2019), and **MRDR-JL** (Guo et al., 2021). In addition, **Naive Bayes with Laplace smoothing** and **logistic regression** are used to establish the propensity model respectively.

**Experimental protocols and details.** The following four metrics are used simultaneously in the evaluation of debiasing performance: *MSE, AUC, NDCG@5,* and *NDCG@10*. All the experiments are implemented on PyTorch with Adam as the optimizer[5]. We tune the learning rate in $\{0.005, 0.01, 0.05, 0.1\}$, weight decay in $\{1e-6, 5e-6, \dots, 5e-3, 1e-2\}$, constrain parameter eta in $\{50, 100, 150, 200\}$ for **Coat** and $\{500, 1000, 1500, 2000\}$ for **Yahoo! R3**, and batch size in $\{128, 256, 512, 1024, 2048\}$ for **Coat** and $\{1024, 2048, 4096, 8192, 16384\}$ for **Yahoo! R3**. In addition, for the Laplacian smooth parameter in Naive Bayes model, the initial value is set to 0 and the learning rate is tuned in $\{5, 10, 15, 20\}$ for **Coat** and in $\{50, 100, 150, 200\}$ for **Yahoo! R3**.

## 5.2 PERFORMANCE COMPARISON (RQ1)

Table 1 summarizes the performance of the proposed Stable-DR and Stable-MRDR methods compared with previous methods. First, the causally-inspired methods perform better than the base model, verifying the necessity of handling the selection bias in rating prediction. For previous meth-

---

[5]For all experiments, we use NVIDIA GeForce RTX 3090 as the computing resource.

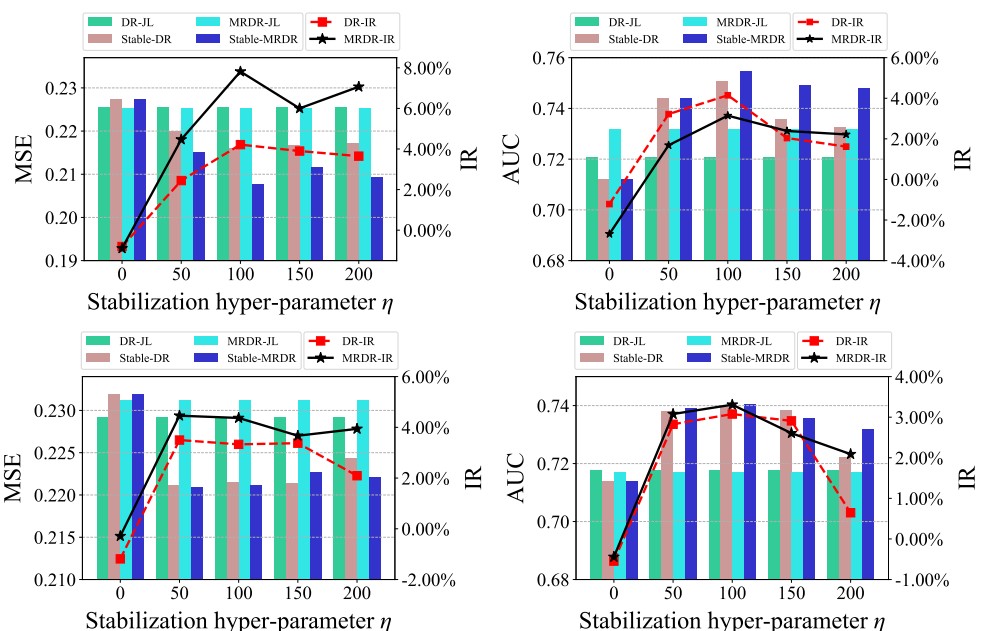

Figure 2: MSE, AUC and Increasing Ratio (IR) of Stable-DR and Stable-MRDR comparing with two baseline algorithms DR-JL and MRDR-JL in two different propensity model setting: Naive Bayes with Laplace smoothing (Top) and logistic regression (Bottom) respectively.

ods, SNIPS, CVIB and DR demonstrate competitive performance. Second, the proposed Stable-DR and Stable-MRDR have the best performance in all four metrics. On one hand, our methods outperform SNIPS, attributed to the inclusion of the propensity model in the training process, as well as the boundedness and double robustness of SDR. On the other hand, our methods outperform DR-JL and MRDR-JL, attributed to the stabilization constraint introduced in the training of the propensity model. This further demonstrates the benefit of cycle learning, in which the propensity model is acted as the mediation between the imputation and prediction model during the training process, rather than updating the prediction model from the imputation model directly.

## 5.3 ABLATION AND PARAMETER SENSITIVITY STUDY (RQ2, RQ3)

The debiasing performance under different stabilization constraint strength and propensity models is shown in Figure 2. First, the proposed Stable-DR and Stable-MRDR outperform DR-JL and MRDR-JL, when either Naive Bayes with Laplace smoothing or logistic regression is used as propensity models. It indicates that our methods have better debiasing ability in both the feature containing and collaborative filtering scenarios. Second, when the strength of the stabilization constraint is zero, our method performs similarly to SNIPS and slightly worse than the DR-JL and MRDR-JL, which indicates that simply using cross-entropy loss to update the propensity model is not effective in improving the model performance. However, as the strength of the stabilization constraint increases, Stable-DR and Stable-MRDR using cycle learning have a stable and significant improvement compared to DR-JL and MRDR-JL. Our methods achieve the optimal performance at the appropriate constraint strength, which can be interpreted as simultaneous consideration of accuracy and stability to ensure boundedness and double robustness of SDR.

## 6 CONCLUSION

In this paper, we propose an SDR estimator for data MNAR that maintains double robustness and improves the stability of DR in the following three aspects: first, we show that SDR has a weaker extrapolation dependence than DR and can result in more stable and accurate predictions in the presence of MNAR effects. Next, through theoretical analysis, we show that the proposed SDR has bounded bias, variance, and generalization error bounds under inaccurate imputed errors and arbitrarily small estimated propensities, while DR does not. Finally, we propose a novel learning approach for SDR that updates the imputation, propensity, and prediction models cyclically, achieving more stable and accurate predictions. Extensive experiments show that our approach significantly outperforms the existing methods in terms of both convergence and prediction accuracy.

ETHICS STATEMENT

This work is mostly theoretical and experiments are based on synthetic and public datasets. We claim that this work does not present any foreseeable negative social impact.

REPRODUCIBILITY STATEMENT

Code is provided in Supplementary Materials to reproduce the experimental results.

ACKNOWLEDGMENTS

The work was supported by the National Key R&D Program of China under Grant No. 2019YFB1705601.

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

## APPENDIX

Throughout, following existing studies (Schnabel et al., 2016; Wang et al., 2019; Guo et al., 2021; Dai et al., 2022), we assume that the indicator matrix $\mathcal{O}$ contains independent random variables and each $o_{u,i}$ follows a Bernoulli distribution with probability $p_{u,i}$.

## A  PROOF OF THEOREMS

### A.1  PROOF OF THEOREM 1

*Proof of Theorem 1.* To demonstrate the double robustness of the SDR, first note that $P\left(\lim_{|\mathcal{D}| \to \infty} \mathcal{E}_{SDR} = \mathcal{L}_{ideal}\right) = 1$ if the learned propensities are accurate (Swaminathan & Joachims, 2015), since $|\mathcal{D}|^{-1} \sum_{(u,i) \in \mathcal{D}} o_{u,i}/\hat{p}_{u,i}$ converges to 1 almost surely as $|\mathcal{D}|$ goes to infinity and IPS is unbiased. Besides, the constraint (1) is constructed to ensure the unbiasedness of $\mathcal{E}_{SDR}$ if the error imputation model is correctly specified. In fact, $\mathcal{E}_{SDR}$ satisfies

$$\frac{1}{|\mathcal{D}|} \sum_{(u,i) \in \mathcal{D}} \frac{o_{u,i}(e_{u,i} - \mathcal{E}_{SDR})}{\hat{p}_{u,i}} = 0. \tag{4}$$

Combining the constraint (1) and equation (4) gives

$$\frac{1}{|\mathcal{D}|} \sum_{(u,i) \in \mathcal{D}} \left[ \frac{o_{u,i}(e_{u,i} - \hat{e}_{u,i})}{\hat{p}_{u,i}} + \frac{o_{u,i}(\hat{\mathcal{E}} - \mathcal{E}_{SDR})}{\hat{p}_{u,i}} \right] = 0,$$

where the first term equals to 0 when the imputation model is correctly specified, it implies that $\mathcal{E}_{SDR} = \hat{\mathcal{E}}$, then the unbiasedness of $\mathcal{E}_{SDR}$ follows immediately from the unbiasedness of $\hat{\mathcal{E}}$.

$\square$

### A.2  PROOF OF THEOREM 2

*Proof of Theorem 2.* Equation (3) implies that $\mathcal{E}_{SDR}$ can be expressed as

$$\mathcal{E}_{SDR} = \left[ \frac{1}{|\mathcal{D}|} \sum_{(u,i) \in \mathcal{D}} \frac{o_{u,i}(e_{u,i} - \hat{e}_{u,i} + \hat{\mathcal{E}})}{\hat{p}_{u,i}} \right] \Big/ \left[ \frac{1}{|\mathcal{D}|} \sum_{(u,i) \in \mathcal{D}} \frac{o_{u,i}}{\hat{p}_{u,i}} \right]. \tag{5}$$

For notational simplicity, let

$$w_{u,i} \triangleq o_{u,i}/\hat{p}_{u,i} \quad \text{and} \quad v_{u,i} \triangleq o_{u,i}(e_{u,i} - \hat{e}_{u,i} + \hat{\mathcal{E}})/\hat{p}_{u,i},$$

then $\mathcal{E}_{SDR}$ can be written as a ratio statistic

$$\mathcal{E}_{SDR} = \frac{1}{|\mathcal{D}|} \sum_{(u,i) \in \mathcal{D}} v_{u,i} \Big/ \frac{1}{|\mathcal{D}|} \sum_{(u,i) \in \mathcal{D}} w_{u,i} \triangleq f(\bar{v}, \bar{w}),$$

where $f(v, w) = v/w$, $\bar{v} = |\mathcal{D}|^{-1} \sum_{(u,i) \in \mathcal{D}} v_{u,i}$, and $\bar{w} = |\mathcal{D}|^{-1} \sum_{(u,i) \in \mathcal{D}} w_{u,i}$.

Applying the Taylor expansion around $(\mu_v, \mu_w) \triangleq (\mathbb{E}[\bar{v}], \mathbb{E}[\bar{w}])$ yields that

$$f(\bar{v}, \bar{w}) = f(\mu_v, \mu_w) + f'_v(\mu_v, \mu_w)(\bar{v} - \mu_v) + f'_w(\mu_v, \mu_w)(\bar{w} - \mu_w)$$
$$+ \frac{1}{2} \left\{ f''_{vv}(\mu_v, \mu_w)(\bar{v} - \mu_v)^2 + 2f''_{vw}(\mu_v, \mu_w)(\bar{v} - \mu_v)(\bar{w} - \mu_w) + f''_{ww}(\bar{w} - \mu_w)^2 \right\}$$
$$+ R(\tilde{v}, \tilde{w}),$$

where $R(\tilde{v}, \tilde{w})$ is the remainder term. Note that $f''_{vv}(\mu_v, \mu_w) = 0$, $f''_{vw}(\mu_v, \mu_w) = -1/\mu_w^2$, and $f''_{ww}(\mu_v, \mu_w) = 2\mu_v/\mu_w^3$, then taking an expectation on both sides of the Taylor expansion leads to

$$\mathbb{E}(\bar{v}/\bar{w}) = \frac{\mu_v}{\mu_w} - \frac{\text{Cov}(\bar{v}, \bar{w})}{(\mu_w)^2} + \frac{\text{Var}(\bar{w})\mu_v}{(\mu_w)^3} + \mathbb{E}[R(\tilde{v}, \tilde{w})].$$

By some calculations, we have $\mathrm{Cov}(\bar{v}, \bar{w}) = O(|\mathcal{D}|^{-1})$, $\mathrm{Var}(\bar{w}) = O(|\mathcal{D}|^{-1})$, $\mathbb{E}[R(\tilde{v}, \tilde{w})] = o(|\mathcal{D}|^{-1})$. Thus, the bias of $\mathcal{E}_{SDR}$ is given as

$$\mathrm{Bias}(\mathcal{E}_{SDR}) = \left| \frac{1}{|\mathcal{D}|} \sum_{(u,i)\in\mathcal{D}} \left( \delta_{u,i} - \frac{\sum_{(u,i)\in\mathcal{D}} \delta_{u,i} p_{u,i}/\hat{p}_{u,i}}{\sum_{(u,i)\in\mathcal{D}} p_{u,i}/\hat{p}_{u,i}} \right) \right| + O(|\mathcal{D}|^{-1}).$$

$\square$

### A.3 PROOF OF THEOREM 3

*Proof of Theorem 3.* According to the proof of Theorem 2, we have

$$\mathbb{E}[\bar{v}/\bar{w}] - \mu_v/\mu_w = O(|\mathcal{D}|^{-1}). \tag{6}$$

Then the variance of $\mathcal{E}_{SDR}$ can be decomposed into as

$$\begin{aligned}
\mathrm{Var}(\mathcal{E}_{SDR}) = \mathrm{Var}(\bar{v}/\bar{w}) &= \mathbb{E}\left[ \{\bar{v}/\bar{w} - \mathbb{E}(\bar{v}/\bar{w})\}^2 \right] \\
&= \mathbb{E}\left[ \{\bar{v}/\bar{w} - \mu_v/\mu_w\}^2 - 2O(|\mathcal{D}|^{-1}) \cdot \{\bar{v}/\bar{w} - \mu_v/\mu_w\} + O(|\mathcal{D}|^{-2}) \right], \\
&= \mathcal{V}_1 + \mathcal{V}_2 + O(|\mathcal{D}|^{-2}),
\end{aligned}$$

where $\mathcal{V}_1 \triangleq \mathbb{E}[\{\bar{v}/\bar{w} - \mu_v/\mu_w\}^2]$, $\mathcal{V}_2 \triangleq -2O(|\mathcal{D}|^{-1}) \cdot [\mathbb{E}(\bar{v}/\bar{w}) - \mu_v/\mu_w]$. Equation (6) implies that $\mathcal{V}_2 = O(|\mathcal{D}|^{-2})$.

Denote $f(v, w) = v/w$, and apply delta method around $(\mu_v, \mu_w) \triangleq (\mathbb{E}[\bar{v}], \mathbb{E}[\bar{w}])$ to calculate $\mathcal{V}_1$ yields that

$$\begin{aligned}
\mathcal{V}_1 &= \mathbb{E}\left\{ \left[ f(\mu_v, \mu_w) + f'_v(\mu_v, \mu_w)(\bar{v} - \mu_v) + f'_w(\mu_v, \mu_w)(\bar{w} - \mu_w) + O_p(|\mathcal{D}|^{-1}) - f(\mu_v, \mu_w) \right]^2 \right\} \\
&= \mathbb{E}\left\{ \left[ f'_v(\mu_v, \mu_w)(\bar{v} - \mu_v) + f'_w(\mu_v, \mu_w)(\bar{w} - \mu_w) + O_p(|\mathcal{D}|^{-1}) \right]^2 \right\} \\
&= \mathbb{E}\left\{ f'^2_v(\mu_v, \mu_w)(\bar{v} - \mu_v)^2 + 2f'_v(\mu_v, \mu_w)(\bar{v} - \mu_v) f'_w(\mu_v, \mu_w)(\bar{w} - \mu_w) \right. \\
&\quad \left. + f'^2_w(\mu_v, \mu_w)(\bar{w} - \mu_w)^2 \right\} + O(|\mathcal{D}|^{-2}) \\
&= f'^2_v(\mu_v, \mu_w) \mathrm{Var}(\bar{v}) + 2f'_v(\mu_v, \mu_w) f'_w(\mu_v, \mu_w) \mathrm{Cov}(\bar{v}, \bar{w}) + f'^2_w(\mu_v, \mu_w) \mathrm{Var}(\bar{w}) + O(|\mathcal{D}|^{-2})
\end{aligned}$$

Note that $f'_v(\mu_v, \mu_w) = 1/\mu_w$ and $f'_w(\mu_v, \mu_w) = -\mu_v/\mu_w^2$. Then we have

$$\begin{aligned}
\mathcal{V}_1 &= \frac{1}{(\mu_w)^2} \mathrm{Var}(\bar{v}) + 2\frac{-\mu_v}{(\mu_w)^3} \mathrm{Cov}(\bar{v}, \bar{w}) + \frac{(\mu_v)^2}{(\mu_w)^4} \mathrm{Var}(\bar{w}) + O(|\mathcal{D}|^{-2}) \\
&= \frac{(\mu_v)^2}{(\mu_w)^2} \left[ \frac{\mathrm{Var}(\bar{v})}{(\mu_v)^2} - 2\frac{\mathrm{Cov}(\bar{v}, \bar{w})}{\mu_v \mu_w} + \frac{\mathrm{Var}(\bar{w})}{(\mu_w)^2} \right] + O(|\mathcal{D}|^{-2}) \\
&= \frac{\mathbb{E}\left( \bar{v} - \frac{\mu_v}{\mu_w}\bar{w} \right)^2}{\mu_w^2} + O(|\mathcal{D}|^{-2}) \\
&= \frac{\sum_{(u,i)} p_{u,i}(1 - p_{u,i}) h_{u,i}^2/\hat{p}_{u,i}^2}{\left( \sum_{(u,i)} p_{u,i}/\hat{p}_{u,i} \right)^2} + O(|\mathcal{D}|^{-2}),
\end{aligned}$$

where $h_{u,i} = (e_{u,i} - \hat{e}_{u,i}) - \sum_{(u,i)} \{p_{u,i}(e_{u,i} - \hat{e}_{u,i})/\hat{p}_{u,i}\}/\sum_{(u,i)} \{p_{u,i}/\hat{p}_{u,i}\}$ is a bounded difference between $e_{u,i} - \hat{e}_{u,i}$ and its weighted average. The conclusion that the SDR variance is bounded for any propensities is given directly by the self-normalized form of SDR, i.e., the bounded range of SDR is $[\delta_{\min}, \delta_{\max}]$.

$\square$

A.4 PROOF OF THEOREM 4

*Proof of Theorem 4.* The McDiarmid's inequality states that for independent bounded random variables $X_1, X_2, \ldots X_n$, where $X_i \in \mathcal{X}_i$ for all $i$ and a mapping $f : \mathcal{X}_1 \times \mathcal{X}_2 \times \cdots \times \mathcal{X}_n \rightarrow \mathbb{R}$. Assume there exist constant $c_1, c_2, \ldots, c_n$ such that for all $i$,

$$\sup_{x_1, \cdots, x_{i-1}, x_i, x'_i, x_{i+1}, \cdots, x_n} |f(x_1, \ldots, x_{i-1}, x_i, x_{i+1}, \cdots, x_n) - f(x_1, \ldots, x_{i-1}, x'_i, x_{i+1}, \cdots, x_n)| \le c_i.$$

Then, for any $\epsilon > 0$,

$$\mathbb{P}(|f(X_1, X_2, \cdots, X_n) - \mathbb{E}[f(X_1, X_2, \cdots, X_n)]| \ge \epsilon) \le 2 \exp\left(-\frac{2\epsilon^2}{\sum_{i=1}^n c_i^2}\right).$$

In fact, equation (5) implies that the SDR estimator can be written as

$$\mathcal{E}_{SDR} = \sum_{(u,i) \in \mathcal{D}} \frac{o_{u,i}(e_{u,i} - \hat{e}_{u,i})}{\hat{p}_{u,i}} \Big/ \sum_{(u,i) \in \mathcal{D}} \frac{o_{u,i}}{\hat{p}_{u,i}} + \hat{\mathcal{E}},$$

denoted as $f(o_{1,1}, \ldots, o_{u,i}, \ldots, o_{U,I})$. Note that

$$\sup_{o_{u,i}, o'_{u,i}} \left| f(o_{1,1}, \ldots, o_{u,i}, \ldots, o_{U,I}) - f(o_{1,1}, \ldots, o'_{u,i} \ldots, o_{U,I}) \right|$$

$$\le \begin{cases} \delta_{\max} - \dfrac{\delta_{u,i}/\hat{p}_{u,i} + \sum_{\mathcal{D} \setminus (u,i)} o_{u,i}/\hat{p}_{u,i}\delta_{\max}}{1/\hat{p}_{u,i} + \sum_{\mathcal{D} \setminus (u,i)} o_{u,i}/\hat{p}_{u,i}}, & \text{if} \quad \delta_{u,i} \le (\delta_{\min} + \delta_{\max})/2, \\[4mm] \dfrac{\sum_{\mathcal{D} \setminus (u,i)} o_{u,i}/\hat{p}_{u,i}\delta_{\min} + \delta_{u,i}/\hat{p}_{u,i}}{\sum_{\mathcal{D} \setminus (u,i)} o_{u,i}/\hat{p}_{u,i} + 1/\hat{p}_{u,i}} - \delta_{\min}, & \text{if} \quad \delta_{u,i} > (\delta_{\min} + \delta_{\max})/2, \end{cases} \tag{7}$$

where $\mathcal{D} \setminus (u,i)$ is the set of $\mathcal{D}$ excluding the element $(u,i)$.

Next, we focus on analyzing the $\sum_{\mathcal{D} \setminus (u,i)} o_{u,i}/\hat{p}_{u,i}$. The Hoeffding's inequality states that for independent bounded random variables $X_1, \ldots, X_n$ that take values in intervals of sizes $\rho_1, \ldots, \rho_n$ with probability 1 and for any $\epsilon > 0$,

$$\mathbb{P}\left(\left|\sum_k X_k - \mathbb{E}(\sum_k X_k)\right| \ge \epsilon\right) \le 2 \exp\left(\frac{-2\epsilon^2}{\sum_k \rho_k^2}\right).$$

For $\sum_{\mathcal{D} \setminus (u,i)} o_{u,i}/\hat{p}_{u,i}$, we have

$$\mathbb{P}\left(\left|\sum_{\mathcal{D} \setminus (u,i)} o_{u,i}/\hat{p}_{u,i} - \sum_{\mathcal{D} \setminus (u,i)} p_{u,i}/\hat{p}_{u,i}\right| \ge \epsilon\right) \le 2 \exp\left(\frac{-2\epsilon^2}{\sum_{\mathcal{D} \setminus (u,i)} 1/\hat{p}_{u,i}^2}\right),$$

Setting the last term equals to $\eta/2$, and solving for $\epsilon$ gives that with probability at least $1 - \eta/2$, the following inequality holds

$$\left|\sum_{\mathcal{D} \setminus (u,i)} o_{u,i}/\hat{p}_{u,i} - \sum_{\mathcal{D} \setminus (u,i)} p_{u,i}/\hat{p}_{u,i}\right| \le \sqrt{\frac{1}{2}\log\frac{4}{\eta} \sum_{\mathcal{D} \setminus (u,i)} \frac{1}{\hat{p}_{u,i}^2}} \triangleq \epsilon'. \tag{8}$$

Therefore, combining (7) and (8) yields that with probability at least $1 - \eta/2$,

$$\sup_{o_{1,1}, \ldots, o_{u,i}, o'_{u,i}, \ldots, o_{U,I}} \left| f(o_{1,1}, \ldots, o_{u,i}, \ldots, o_{U,I}) - f(o_{1,1}, \ldots, o'_{u,i} \ldots, o_{U,I}) \right|$$

$$\le \begin{cases} \delta_{\max} - \dfrac{\delta_{u,i}/\hat{p}_{u,i} + (\sum_{\mathcal{D} \setminus (u,i)} p_{u,i}/\hat{p}_{u,i} - \epsilon')\delta_{\max}}{1/\hat{p}_{u,i} + (\sum_{\mathcal{D} \setminus (u,i)} p_{u,i}/\hat{p}_{u,i} - \epsilon')}, & \text{if} \quad \delta_{u,i} \le (\delta_{\min} + \delta_{\max})/2, \\[4mm] \dfrac{(\sum_{\mathcal{D} \setminus (u,i)} p_{u,i}/\hat{p}_{u,i} - \epsilon')\delta_{\min} + \delta_{u,i}/\hat{p}_{u,i}}{(\sum_{\mathcal{D} \setminus (u,i)} p_{u,i}/\hat{p}_{u,i} - \epsilon') + 1/\hat{p}_{u,i}} - \delta_{\min}, & \text{if} \quad \delta_{u,i} > (\delta_{\min} + \delta_{\max})/2, \end{cases}$$

$$\le \begin{cases} (\delta_{\max} - \delta_{u,i})/\{1 + \hat{p}_{u,i}(\sum_{\mathcal{D} \setminus (u,i)} p_{u,i}/\hat{p}_{u,i} - \epsilon')\}, & \text{if} \quad \delta_{u,i} \le (\delta_{\min} + \delta_{\max})/2, \\[2mm] (\delta_{u,i} - \delta_{\min})/\{1 + \hat{p}_{u,i}(\sum_{\mathcal{D} \setminus (u,i)} p_{u,i}/\hat{p}_{u,i} - \epsilon')\}, & \text{if} \quad \delta_{u,i} > (\delta_{\min} + \delta_{\max})/2, \end{cases}$$

where $\delta_{u,i} = e_{u,i} - \hat{e}_{u,i}$ is the error deviation, $\delta_{\min} = \min_{(u,i)\in\mathcal{D}} \delta_{u,i}$, and $\delta_{\max} = \max_{(u,i)\in\mathcal{D}} \delta_{u,i}$. Invoking McDiarmid's inequality leads to that

$$
\mathbb{P}\left(|\mathcal{E}_{SDR} - \mathbb{E}_{\mathcal{O}}(\mathcal{E}_{SDR})| \geq \epsilon\right)
$$

$$
\leq 2\exp\Bigg\{-2\epsilon^2 \Bigg/ \Bigg( \sum_{(u,i):\delta_{u,i}\leq\frac{\delta_{\min}+\delta_{\max}}{2}} \frac{(\delta_{\max} - \delta_{u,i})^2}{\{1+\hat{p}_{u,i}(\sum_{\mathcal{D}-(u,i)} p_{u,i}/\hat{p}_{u,i} - \epsilon')\}^2}
$$

$$
+ \sum_{(u,i):\delta_{u,i}>\frac{\delta_{\min}+\delta_{\max}}{2}} \frac{(\delta_{u,i} - \delta_{\min})^2}{\{1+\hat{p}_{u,i}(\sum_{\mathcal{D}-(u,i)} p_{u,i}/\hat{p}_{u,i} - \epsilon')\}^2} \Bigg)\Bigg\}
$$

$$
\leq 2\exp\left(\frac{-2\epsilon^2}{\sum_{(u,i)}\{(\delta_{\max} - \delta_{u,i})^2 + (\delta_{u,i} - \delta_{\min})^2\}/\{1+\hat{p}_{u,i}(\sum_{\mathcal{D}-(u,i)} p_{u,i}/\hat{p}_{u,i} - \epsilon')\}^2}\right)
$$

Setting the last term equals to $\eta/2$, and solving for $\epsilon$ complete the proof.

$\square$

## A.5 GENERALIZATION BOUND UNDER INACCURATE MODELS

**Theorem 5** (Generalization Bound under Inaccurate Models). *For any finite hypothesis space of predictions $\mathcal{H} = \{\hat{\mathbf{Y}}^1, \ldots, \hat{\mathbf{Y}}^{|\mathcal{H}|}\}$, with probability $1 - \eta$, the true risk $R(\hat{\mathbf{Y}}^\dagger)$ deviates from the SDR estimator with imputed errors $\hat{e}_{u,i}$ and learned propensities $\hat{p}_{u,i}$ satisfying the stabilization constraint 1 is bounded by*

$$
R(\hat{\mathbf{Y}}^\dagger) \leq \hat{\mathcal{E}}_{SDR}(\hat{\mathbf{Y}}^\dagger) + \underbrace{\left| \frac{1}{|\mathcal{D}|}\sum_{(u,i)\in\mathcal{D}} \delta_{u,i}^\dagger - \frac{\sum_{(u,i)\in\mathcal{D}} \delta_{u,i}^\dagger p_{u,i}/\hat{p}_{u,i}}{\sum_{(u,i)\in\mathcal{D}} p_{u,i}/\hat{p}_{u,i}} \right|}_{\text{Bias Term}}
$$

$$
+ \underbrace{\sqrt{\frac{1}{2}\log\left(\frac{4|\mathcal{H}|}{\eta}\right) \sum_{(u,i)\in\mathcal{D}} \frac{(\delta_{\max} - \delta_{u,i}^\S)^2 + (\delta_{u,i}^\S - \delta_{\min})^2}{\{1+\hat{p}_{u,i}(\sum_{\mathcal{D}\setminus(u,i)} p_{u,i}/\hat{p}_{u,i} - \epsilon')\}^2}}}_{\text{Variance Term}}
$$

*where $\delta_{u,i}^\S$ is the error deviation corresponding to the prediction model*

$$
\hat{\mathbf{Y}}^\S = \operatorname{argmax}_{\hat{\mathbf{Y}}^h \in \mathcal{H}} \sum_{(u,i)\in\mathcal{D}} \frac{(\delta_{\max} - \delta_{u,i}^\S)^2 + (\delta_{u,i}^\S - \delta_{\min})^2}{\{1+\hat{p}_{u,i}(\sum_{\mathcal{D}\setminus(u,i)} p_{u,i}/\hat{p}_{u,i} - \epsilon')\}^2}.
$$

*Proof of Theorem 5.* Proof. Theorem 4 shows that for all predictions $\hat{\mathbf{Y}}^h \in \mathcal{H}$, we have

$$
P\left(\left|\mathcal{E}_{SDR}(\hat{\mathbf{Y}}^h) - \mathbb{E}[\mathcal{E}_{SDR}(\hat{\mathbf{Y}}^h)]\right| \geq \epsilon\right)
$$

$$
\leq 2\exp\left(\frac{-2\epsilon^2}{\sum_{(u,i)}\{(\delta_{\max} - \delta_{u,i}^h)^2 + (\delta_{u,i}^h - \delta_{\min})^2\}/\{1+\hat{p}_{u,i}\}^2}\right)
$$

McDiarmid's inequality and union bound ensures the following uniform convergence results:

$$P\left(\left|\mathcal{E}_{SDR}(\hat{\mathbf{Y}}^\dagger) - \mathbb{E}[\mathcal{E}_{SDR}(\hat{\mathbf{Y}}^\dagger)]\right| \le \epsilon\right) \ge 1 - \eta$$

$$\Leftarrow P\left(\max_{\hat{\mathbf{Y}}^h \in \mathcal{H}} \left|\mathcal{E}_{SDR}(\hat{\mathbf{Y}}^h) - \mathbb{E}[\mathcal{E}_{SDR}(\hat{\mathbf{Y}}^h)]\right| \le \epsilon\right) \ge 1 - \eta$$

$$\Leftrightarrow P\left(\bigvee_{\hat{\mathbf{Y}}_i \in \mathcal{H}} \left|\mathcal{E}_{SDR}(\hat{\mathbf{Y}}^h) - \mathbb{E}[\mathcal{E}_{SDR}(\hat{\mathbf{Y}}^h)]\right| \ge \epsilon\right) < \eta$$

$$\Leftarrow \sum_{h=1}^{|\mathcal{H}|} P\left(\left|\mathcal{E}_{SDR}(\hat{\mathbf{Y}}^h) - \mathbb{E}[\mathcal{E}_{SDR}(\hat{\mathbf{Y}}^h)]\right| \ge \epsilon\right) < \eta$$

$$\Leftarrow \sum_{h=1}^{|\mathcal{H}|} 2\exp\left(\frac{-2\epsilon^2}{\sum_{(u,i)}\{(\delta_{\max} - \delta_{u,i}^h)^2 + (\delta_{u,i}^h - \delta_{\min})^2\}/\{1 + \hat{p}_{u,i}(\sum_{\mathcal{D}-(u,i)} p_{u,i}/\hat{p}_{u,i} - \epsilon')\}^2}\right) < \eta$$

$$\Leftarrow |\mathcal{H}| \cdot 2\exp\left(\frac{-2\epsilon^2}{\sum_{(u,i)}\{(\delta_{\max} - \delta_{u,i}^\S)^2 + (\delta_{u,i}^\S - \delta_{\min})^2\}/\{1 + \hat{p}_{u,i}(\sum_{\mathcal{D}-(u,i)} p_{u,i}/\hat{p}_{u,i} - \epsilon')\}^2}\right) < \eta$$

Solving the last inequality for $\epsilon$, it is concluded that, with probability $1 - \eta$, the following inequality holds

$$\mathbb{E}[\mathcal{E}_{SDR}(\hat{\mathbf{Y}}^\dagger)] - \mathcal{E}_{SDR}(\hat{\mathbf{Y}}^\dagger) \le \sqrt{\frac{1}{2}\log\left(\frac{4|\mathcal{H}|}{\eta}\right)\sum_{(u,i)\in\mathcal{D}}\frac{(\delta_{\max} - \delta_{u,i}^\S)^2 + (\delta_{u,i}^\S - \delta_{\min})^2}{\{1 + \hat{p}_{u,i}(\sum_{\mathcal{D}\backslash(u,i)} p_{u,i}/\hat{p}_{u,i} - \epsilon')\}^2}}.$$

Theorem 2 shows that for the optimal prediction model $\hat{\mathbf{Y}}^\dagger$, the following inequality holds

$$R(\hat{\mathbf{Y}}^\dagger) - \mathbb{E}[\mathcal{E}_{SDR}(\hat{\mathbf{Y}}^\dagger)] \le \left|\frac{1}{|\mathcal{D}|}\sum_{(u,i)\in\mathcal{D}}\delta_{u,i}^\dagger - \frac{\sum_{(u,i)\in\mathcal{D}}\delta_{u,i}^\dagger p_{u,i}/\hat{p}_{u,i}}{\sum_{(u,i)\in\mathcal{D}} p_{u,i}/\hat{p}_{u,i}}\right|.$$

The stated results can be obtained by adding the two inequalities above.

$\square$

## B  FURTHER THEORETICAL ANALYSIS OF SDR

Without loss of generality, we assume

$$\frac{1}{|\mathcal{D}|}\sum_{(u,i)\in\mathcal{D}}\frac{o_{u,i}}{\hat{p}_{u,i}}(\hat{e}_{u,i} - \hat{\mathcal{E}}) = \lambda, \quad \lambda \ne 0.$$

In this case, the learned propensities must be inaccurate; otherwise, the constraint (1) holds naturally as the same size increases. Thus, if the imputed errors are accurate, then $\mathcal{L}_{ideal} = \hat{\mathcal{E}}$. By a exactly same arguments of equation (3), we have

$$\frac{1}{|\mathcal{D}|}\sum_{(u,i)\in\mathcal{D}}\frac{o_{u,i}}{\hat{p}_{u,i}}(\hat{\mathcal{E}} - \mathcal{E}_{SDR}) = \lambda,$$

which implies that

$$\mathcal{E}_{SDR} = \mathcal{L}_{ideal} - \lambda\Big/\frac{1}{|\mathcal{D}|}\sum_{(u,i)\in\mathcal{D}}\frac{o_{u,i}}{\hat{p}_{u,i}}.$$

This means that the degree of violation of constraint (1) determines the size of the bias of SDR.

Furthermore, we can compute the bias, variance, tail bound, and generalization error bound of SDR. Specifically, if both the learned propensities and imputed errors are inaccurate, constraint (3) does not hold either. Then the bias of SDR is

$$\text{Bias}(\mathcal{E}_{SDR}) = \left|\frac{1}{|\mathcal{D}|}\sum_{(u,i)\in\mathcal{D}}\left(e_{u,i} - \frac{\sum_{(u,i)\in\mathcal{D}} e_{u,i}p_{u,i}/\hat{p}_{u,i}}{\sum_{(u,i)\in\mathcal{D}} p_{u,i}/\hat{p}_{u,i}}\right)\right| + O(|\mathcal{D}|^{-1}),$$

the variance of SDR becomes

$$\text{Var}\left(\mathcal{E}_{SDR}\right) = \frac{\sum_{(u,i)} p_{u,i}(1 - p_{u,i})\tilde{h}_{u,i}^2/\hat{p}_{u,i}^2}{\left(\sum_{(u,i)} p_{u,i}/\hat{p}_{u,i}\right)^2} + O(|\mathcal{D}|^{-2}),$$

where $\tilde{h}_{u,i} = e_{u,i} - \sum_{(u,i)\in\mathcal{D}}\{p_{u,i}e_{u,i}/\hat{p}_{u,i}\}\big/ \sum_{(u,i)\in\mathcal{D}}\{p_{u,i}/\hat{p}_{u,i}\}$. The tail bound of SDR is given as

$$|\mathcal{E}_{SDR} - \mathbb{E}_{\mathcal{O}}(\mathcal{E}_{SDR})| \leq \sqrt{\frac{1}{2}\log\left(\frac{4}{\eta}\right)\sum_{(u,i)\in\mathcal{D}}\frac{(e_{\max} - e_{u,i})^2 + (e_{u,i} - e_{\min})^2}{\{1 + \hat{p}_{u,i}(\sum_{\mathcal{D}\backslash(u,i)} p_{u,i}/\hat{p}_{u,i} - \epsilon')\}^2}},$$

where $\delta_{\min} = \min_{(u,i)\in\mathcal{D}}e_{u,i}$, $\delta_{\max} = \max_{(u,i)\in\mathcal{D}}e_{u,i}$, $\epsilon' = \sqrt{\log(4/\eta)/2 \cdot \sum_{\mathcal{D}\backslash(u,i)} 1/\hat{p}_{u,i}^2}$, and $\mathcal{D} \backslash (u, i)$ is the set of $\mathcal{D}$ excluding the element $(u, i)$. In addition, we can derive the generation error bound of SDR. Given a finite hypothesis space $\mathcal{H}$ of the prediction model, then for any a prediction model $h \in \mathcal{H}$, with probability $1 - \eta$, the true risk $R(h)$ deviates from the SDR estimator is bounded by

$$R(h) \leq \hat{\mathcal{E}}_{SDR}(h) + \text{Bias}(\mathcal{E}_{SDR}) + \sqrt{\frac{1}{2}\log(\frac{4|\mathcal{H}|}{\eta})\sum_{(u,i)\in D}\frac{(e_{\max} - e_{u,i}^\S)^2 + (e_{u,i}^\S - e_{\min})^2}{\{1 + \hat{p}_{u,i}(\sum_{\mathcal{D}\backslash(u,i)} p_{u,i}/\hat{p}_{u,i} - \epsilon')\}^2}},$$

where $e_{u,i}^\S$ is the error deviation corresponding to the prediction model

$$h^\S = \arg\max_{h\in\mathcal{H}} \sum_{(u,i)\in D}\frac{(e_{max} - e_{u,i}^\S)^2 + (e_{u,i}^\S - e_{min})^2}{\{1 + \hat{p}_{u,i}(\sum_{D-(u,i)} p_{u,i}/\hat{p}_{u,i} - \epsilon')\}^2}.$$

