# OpenReview forum: "StableDR: Stabilized Doubly Robust Learning for Recommendation on Data Missing Not at Random"
_ICLR.cc/2023/Conference — ICLR 2023 poster_

### Official Review · Reviewer_HkXH · 2022-10-23

**Confidence:** 4
**Correctness:** 3
**Technical Novelty And Significance:** 3
**Empirical Novelty And Significance:** 3
**Recommendation:** 8

**Clarity, Quality, Novelty And Reproducibility:**

- The paper is very well-written, and it's very easy to understand.
- The idea presented is novel, and the appendix has detailed proof of the tail bounds used in the paper.
- Authors have provided the source code as supplementary material.

**Strength And Weaknesses:**

- The paper is well-written and easy to understand. The related works are covered in detail and are up-to-date with the latest work in learning from MNAR data.
- Theorm 4 (Tail Bound of SDR) is convincing to show that the proposed method is robust to small propensities in the data.
- Given that the overall learning is done in three stages, I was expecting a relatively larger difference in training time, but Table 2 suggests the proposed method has a runtime similar to existing methods, which is impressive, given the stability guarantees of the methods.

Some weaknesses of the paper:

- SDR is asymptotically unbiased (Theorm 1), whereas baselines like IPS are unbiased for any given data size. This means, for smaller datasets, there is no guarantee of unbiasedness. Although the experiments on small datasets like Coat demonstrate superior performance to baselines, there's still there's no guarantee of unbiasedness when the dataset is small. Any comments on that?
- Since the SDR estimator is similar in form to the SNIPS estimator, it would be interesting to compare the bias-variance of both estimators. Since SNIPS performs some kind of normalization, it should also be robust to small propensities.

**Summary Of The Paper:**

The authors propose a new Stable Doubly-Robust (SDR) method for learning a recommender system from Missing Not At Random (MNAR) feedback data. As compared to previous efforts on DR learning from MNAR data, current work provides theoretical guarantees of stability, specifically against very small propensities. Previous works like Inverse Propensity Scoring (IPS), and DR are prone to instability when certain items have very low propensities in the log data. Due to the inverse term, a very low propensity can cause very high variance.

The authors also propose a 'pipeline' for learning, where the imputation model, propensity model, and SDR model are trained serially/cyclically, to ensure stability in the learning process. Theoretical bounds of SDR learning guarantee robustness against small propensities, which is a downside of the existing methods, where small propensities can lead to larger generalization errors (as shown on Page 6).

**Summary Of The Review:**

- The authors propose a new Stable Doubly-Robust (SDR) method for learning a recommender system from Missing Not At Random (MNAR) feedback data.
- The method is proven to be robust under the presence of small propensities, a setting in which the existing methods might fail.
- Empirical results on popular MNAR benchmarks (Coat and Yahoo! R3) demonstrate the efficacy of the proposed method against strong baselines.

-- Response to authors ---

The authors have addressed all the comments and some concerns I had initially. Overall I think this is a very interesting paper with extensive theoretical analysis of the generalization properties of the method. I think this paper would be very relevant to the ML & RecSys community.

---

> ### Author Response · Authors · 2022-11-17
> **Response to Reviewer HkXH (Part 1 of 2)**
>
> Thank you for the helpful and constructive comments. We addressed all your concerns in our revised version. Specifically, we discussed in detail the meaning of "asymptotic unbiasedness" and discussed the difference between the proposed SDR and SNIPS both theoretically and empirically. We added extensive numerical simulations in Appendix D and real-world experiments in Appendix C, showing that the proposed SDR significantly outperforms IPS, SNIPS, and DR on eight semi-synthetic datasets with different sample sizes and sparsity. Please kindly find our responses below.
>
> > _**Q1**: SDR is asymptotically unbiased (Theorm 1), whereas baselines like IPS are unbiased for any given data size. This means, for smaller datasets, there is no guarantee of unbiasedness. Although the experiments on small datasets like Coat demonstrate superior performance to baselines, there's still there's no guarantee of unbiasedness when the dataset is small. Any comments on that?_
>
> **A1**: Thanks for your comments. We would like to clarify that the requirements of SDR to achieve unbiasedness has no essential difference comparing to the existing methods, e.g., IPS and DR.  Strictly speaking, both $\hat p_{u,i}$ and $\hat e_{u,i}$ **cannot** be estimated accurately for $p_{u,i}$ and $e_{u,i}$ using fixed sample size, unless the sample size goes to infinity.  Though IPS estimator is unbiased if $\hat p_{u,i}=p_{u,i}$ for all user-item pairs $(u, i)\in \mathcal{O}$, the condition "$\hat p_{u,i}=p_{u,i}$" is so strong that can only hold as the sample size goes to infinity, i.e., in the sense of "asymptotically unbiased", instead of "any given data size".
>
> Formally, in statistics, beside the IPS and DR estimators, all "unbiasedness" is expressed in an "asymptotically unbiased" sense [1, 2, 3, 4].  In recommender systems' community, we usually ignore the notation "asymptotically unbiased" for simplification.
>
> >_**Q2**: Since the SDR estimator is similar in form to the SNIPS estimator, it would be interesting to compare the bias-variance of both estimators. Since SNIPS performs some kind of normalization, it should also be robust to small propensities._
>
> **A2**:  Thanks for your comments.
>
> **(a) Bias-variance comparison on SNIPS and SDR estimators.**
>
> Theorems 2 and 3 show that
> $$
> \mbox{Bias}(L_{SDR}) = \Biggl |\frac{1}{|\mathcal{D}|} \sum_{(u,i)\in \mathcal{D}} \left ( \delta_{u, i}-\frac{\sum_{(u,i)\in  \mathcal{D}}{ \delta_{u, i}p_{u,i}}/\hat p_{u,i}}{\sum_{(u,i)\in \mathcal{D}}{p_{u,i}}/\hat p_{u,i}} \right ) \Biggr | + O( |\mathcal{D}|^{-1} ),
> $$
>
> $$
> \mbox{Var}\left(L_{SDR}\right)=\frac{\sum_{(u,i) \in \mathcal{D} } p_{u,i}(1-p_{u,i})h^2_{u,i}/\hat p^2_{u,i}}{\left(\sum_{(u,i)\in \mathcal{D}}p_{u,i}/\hat p_{u,i}\right)^2}+O( |\mathcal{D}|^{-2} ),
> $$
> where $\delta_{u,i} = e_{u,i} - \hat e_{u,i}$, $h_{u,i}=\delta_{u, i}-\sum_{(u,i)\in \mathcal{D}}\{p_{u,i}\delta_{u, i}/\hat p_{u,i}\} \Big / \sum_{(u,i)\in \mathcal{D}}\{p_{u,i}/\hat p_{u,i}\}$.
> In addition, when we remove the constraint (3), then SDR estimator reduces to  SNIPS estimator. In Appendix B of the revised version, we show that the bias and variance of the SNIPS estimator and the bias and variance of the SDR estimate have the same form, except that $\delta_{u,i}$ is replaced by $e_{u,i}$. That is,
> $$
>     \
> \mbox{Bias}(L_{SNIPS}) = \Biggl |\frac{1}{|\mathcal{D}|} \sum_{(u,i)\in  \mathcal{D}} \left ( e_{u, i}-\frac{\sum_{(u,i)\in  \mathcal{D}}{ e_{u, i}p_{u,i}}/\hat p_{u,i}}{\sum_{(u,i)\in \mathcal{D}}{p_{u,i}}/\hat p_{u,i}} \right ) \Biggr | + O( |\mathcal{D}|^{-1} ),
> \
> $$
>
> $$
>     \
> \mbox{Var}\left(L_{SNIPS}\right)=\frac{\sum_{(u,i) \in \mathcal{D}} p_{u,i}(1-p_{u,i}) \tilde h^2_{u,i}/\hat p^2_{u,i}}{\left(\sum_{(u,i)\in \mathcal{D}}p_{u,i}/\hat p_{u,i}\right)^2}+O( |\mathcal{D}|^{-2} ),
> \
> $$
> where $\tilde h_{u,i}=e_{u,i} -\sum_{(u,i)\in \mathcal{D}}\{p_{u,i}e_{u,i}/\hat p_{u,i}\} \Big / \sum_{(u,i)\in \mathcal{D}}\{p_{u,i}/\hat p_{u,i}\}$.
>
> Thus, when $|e_{u,i} - \hat e_{u,i} | < e_{u,i}$, that is, $0< \hat e_{u,i} < 2 e_{u,i}$, the bias and variance of SDR estimator tend to smaller than those of SNIPS estimator.
>
> **(b) SDR has double robustness property, while SNIPS does NOT.**
>
> It is well known that SNIPS uses a self-normalized form to guarantee robustness to small propensities. However, as shown in Theorem 1, the proposed SDR is robust to small propensities while also having the double robustness property, that is, it is unbiased  when either learned propensities or imputed errors are accurate.
>  Empirically, doubly robust estimators always outperform "single robust" estimators such as IPS and SNIPS. Thus, combining (a) and (b), the proposed SDR has smaller bias and variance compared to SNIPS when $0< \hat e_{u,i} < 2 e_{u,i}$, as well as double robustness and weaker extrapolation requirements than DR (see the detailed discussion in Section 3.1).

---

> > ### Author Response · Authors · 2022-11-17
> > **Response to Reviewer HkXH (Part 2 of 2)**
> >
> > **(c) Empirical evidence that SDR outperforms SNIPS under varying bias levels.**
> >
> > To fully illustrate the performance of the proposed SDR compared to SNIPS and DR, we added extensive Monte Carlo studies in Appendix D, to show that the SDR has smaller bias and variance compared to previous methods empirically. Following the previous studies [5, 6, 7, 8], we simulated four different selection bias levels using **MovieLens 100K (ML-100K)** and **MovieLens 1M (ML-1M)**, respectively, and named them as: **ML-100K-1, ML-100K-2, ML-100K-3, ML-100K-4**, and  **ML-1M-1, ML-1M-2, ML-1M-3, ML-1M-4** with descending order of the propensities scale. Specifically, we set the true propensity $p_{u, i}\in(0,1)$ for each rating $R_{u, i}$, with $p_{u, i}=p \alpha^{1+0.25\cdot\min (4,  5-R_{u, i})\cdot  k}$, where $p=1$, $\alpha=0.5$, $k\in \\{1, 2, 3, 4\\}$, and $R_{u, i} \in\\{1, 2, 3, 4, 5\\}$.
> > The absolute relative error (RE) is used as the evaluation protocol, which is defined as follows
> > $$
> > \\operatorname{RE} (L_{{estimator}})= \frac{|L_{{ideal }}(\hat{\mathbf{R}})-L_{ {estimator}}(\hat{\mathbf{R}})|}{L_{{ideal}}(\hat{\mathbf{R}})}\,
> > $$
> > where $L_{{estimator}}$ denotes the estimator to be compared. RE evaluates the accuracy of the estimated loss, the smaller the RE, the more accurate the estimation of the ideal loss. More experimental details are provided in Appendix D.1.
> >
> > We performed 100 repeated experiments on **ML-100K** and  **ML-1M** to compare the estimation performance of Naive, IPS, SNIPS, DR and the proposed SDR under different selection bias levels, and the results are shown in Table 10 in Appendix D. We have the following findings: first, overall, SDR performs significantly better than baselines and the variance is smaller compared to both DR and SNIPS, which empirically demonstrates the stability of our method. Second, there is no significant difference in the performance of IPS and SNIPS in all cases, but our proposed SDR estimator outperforms DR estimator at a statistically significant level of 0.001 in all cases. This suggests that our superior performance is not attributed to self-normalize form, but to carefully designed model dependency, which utilizes the propensity model and imputation model simultaneously in a differentiated way. Third, as the selection bias level increases from **ML-100K/1M-1** to **ML-100K/1M-4**, it can be seen that the estimation error of all baseline estimators increases significantly, while SDR does not. This is attributed to the fact that the proposed SDR estimator is more stable to small propensities.
> >
> > ***
> > We hope that we have addressed the reviewer's concerns, and look forward to future engagement about our work.
> > ***
> >
> > **References**
> >
> > [1] Joseph D.Y. Kang and Joseph L. Schafer. Demystifying double robustness: a comparison of alternative strategies for estimating a population mean from incomplete data. Statistical Science, 22:523–539, 2007.
> >
> > [2] Zhiqiang Tan. Comment: understanding OR, PS and DR. Statistical Science, 22:560–568, 2007.
> >
> > [3]  Vermeulen, K. and Vansteelandt, S. (2015). Bias-reduced doubly robust estimation. Journal of the American Statistical Association 110, 1024–1036
> >
> > [4]  Shaun R. Seaman and Stijn Vansteelandt (2018). Introduction to Double Robust Methods for Incomplete Data. Statistical Science, 33:184-197, 2018.
> >
> > [5] Tobias Schnabel, Adith Swaminathan, Ashudeep Singh, Navin Chandak, and Thorsten Joachims. Recommendations as treatments: Debiasing learning and evaluation. ICML, 2016.
> >
> > [6] Xiaojie Wang, Rui Zhang, Yu Sun, and Jianzhong Qi. Doubly robust joint learning for recommendation on data missing not at random. In ICML, 2019.
> >
> > [7] Yuta Saito. Doubly robust estimator for ranking metrics with post-click conversions. In SIGIR, 2020.
> >
> > [8] Siyuan Guo, Lixin Zou, Yiding Liu, Wenwen Ye, Suqi Cheng, Shuaiqiang Wang, Hechang Chen, Dawei Yin, and Yi Chang. Enhanced doubly robust learning for debiasing post-click conversion rate estimation. In SIGIR, 2021.

---

> > ### Comment · Reviewer_HkXH · 2022-11-18
> > **Question to the authors**
> >
> > Thanks for your detailed response, I appreciate the time and efforts the authors have put into the rebuttal. I am still processing the text, meanwhile, I have a quick clarification question for the authors.
> >
> > \textit{Though IPS estimator is unbiased if $\hat p_{u,i}=p_{u,i}$ for all user-item pairs $(u, i)\in \mathcal{O}$, the condition "$\hat p_{u,i}=p_{u,i}$" is so strong that can only hold as the sample size goes to infinity, i.e., in the sense of "asymptotically unbiased", instead of "any given data size".}
> >
> > This is when the propensities are estimated from the log data. For a situation when the exact propensities are known (IMO quite a realistic situation, since the system designer can log the actual action propensities), the unbiasedness of IPS holds for "any given data-size", right?
> > For the setting where the propensities need to be estimated from data, your statement is valid. But in a real-world setting, for ex. in a recommender system, the system designer can log the action probability for an item, given the user context.
> >
> > Any comments on this? Thanks!

---

> > > ### Author Response · Authors · 2022-11-18
> > > **Thanks and we would like to clarify the necessity of estimating the propensities in data missing-not-at-random setting.**
> > >
> > > Thank you for your timely and detailed comments, we apologize for the ambiguity of the original problem formalization, and have revised and submitted our manuscript (please kindly refer to the red text in Section 2.1). We would like to make the following clarifications.
> > >
> > > First, we fully agree that
> > >
> > > > For a situation when the exact propensities are known, the unbiasedness of IPS holds for "any given data-size".
> > >
> > > and
> > >
> > > > For the setting where the propensities need to be estimated from data, our statement is valid.
> > >
> > > However, in contrast to the known propensities of a given recommendation strategy by the system designer as pointed by the reviewer, we focus on the problem of data missing-not-at-random (same as our paper title), due to the presence of selection bias in the collected ratings. Specifically, selection bias happens in explicit feedback data **as users are free to choose which items to rate**, so that the observed ratings are not a representative sample of all ratings. Thus, **the propensities $p_{u, i}$ here should be interpreted as the probability that user $u$ will rate item $i$**, which are decided by the user selective behaviors, and prediction models are trained using these collected biased ratings **without knowing the exact propensities**, regardless of whether the recommendation policy is known or not. A similar setup can be seen from the previous studies [1, 2, 3, 4, 5, 6].
> > >
> > > **References**
> > >
> > > [1] Tobias Schnabel, Adith Swaminathan, Ashudeep Singh, Navin Chandak, and Thorsten Joachims. Recommendations as treatments: Debiasing learning and evaluation. ICML, 2016.
> > >
> > > [2] Xiaojie Wang, Rui Zhang, Yu Sun, and Jianzhong Qi. Doubly robust joint learning for recommendation on data missing not at random. In ICML, 2019.
> > >
> > > [3] Siyuan Guo, Lixin Zou, Yiding Liu, Wenwen Ye, Suqi Cheng, Shuaiqiang Wang, Hechang Chen, Dawei Yin, and Yi Chang. Enhanced doubly robust learning for debiasing post-click conversion rate estimation. In SIGIR, 2021.
> > >
> > > [4] Yuta Saito. Doubly robust estimator for ranking metrics with post-click conversions. In SIGIR, 2020.
> > >
> > > [5] Xiaojie Wang, Rui Zhang, Yu Sun, and Jianzhong Qi. Combating selection biases in recommender systems with a few unbiased ratings. In WSDM, 2021.
> > >
> > > [6] Jiawei Chen, Hande Dong, Yang Qiu, Xiangnan He, Xin Xin, Liang Chen, Guli Lin, and Keping Yang. Autodebias: Learning to debias for recommendation. In SIGIR, 2021.

---

> > > > ### Comment · Reviewer_HkXH · 2022-11-18
> > > > **Reply to authors**
> > > >
> > > > Thank you very much for clarifying. My apologies for misunderstanding the problem setup, I had the other setting in mind, where the propensities are in the control of the system designer.
> > > >
> > > > This makes a lot of sense now. Thanks for your detailed responses and further experiments added to the paper.
> > > >
> > > > Overall I think this is a very interesting paper, and the extensive theoretical analysis can be helpful for future research in this area.

---

> > > > > ### Author Response · Authors · 2022-11-19
> > > > > **Thank you!**
> > > > >
> > > > > We sincerely appreciate the reviewer's careful review and positive comments on our work. Thank you!

---

### Official Review · Reviewer_f6Th · 2022-10-23

**Confidence:** 4
**Correctness:** 3
**Technical Novelty And Significance:** 2
**Empirical Novelty And Significance:** 2
**Recommendation:** 5

**Clarity, Quality, Novelty And Reproducibility:**

This paper is well-presented and it is easy to follow the logic flow. The ideas proposed by the paper are marginally novel, though.

**Strength And Weaknesses:**

W1: The paper argues that existing DR estimators have an unbounded bias, variance, and generalization bound given extremely small propensities, while the proposed StableDR estimator does not. It would be better to formulate this argument formally into some lemma or proposition. To argue that the StableDR estimator is more stable than existing DR estimators, it is crucial to demonstrate that the StableDR estimator has a smaller variance than the DR estimators, both theoretically and empirically.

W2: I am concerned about the novelty of the proposed approach. First, the StableDR estimator has the same form as an existing Self-Normalized Inverse-Propensity-Scoring (SNIPS) estimator. The property of the StableDR estimator having a bounded bias, variance, and generalization bound mostly comes from the SNIPS estimator. Second, it is not clear why the paper proposes to train models in a cyclic order of prediction model, imputation model, and propensity model instead of another cyclic order or joint learning of all three models. It is important to add more explanations, justifications, and theoretical or empirical results regarding this aspect.

W3: Looks like the property of double robustness in Theorem 1 does not have the exact same definition as that in existing DR estimators. It is recommended to call out the difference and provide some comparisons between these different definition of being double robust.

S1: The experimental results are quite extensive. The paper uses two representative datasets for recommendation debiasing. The paper compares the proposed approach against a wide variety of existing approaches under various metrics. The paper also present some ablation studies to understand the impacts of different components of the proposed approach.

S2: The presentation of the paper is good and the references by the paper are sufficient.

**Summary Of The Paper:**

Existing doubly robust (DR) estimators have multiple limitations, e.g, having an unbounded bias, variance, and generalization bound when propensities are extremely small. To address these limitations, this paper proposes a stabilized doubly robust (StableDR) estimator which has a weaker reliance on extrapolation than the existing DR estimators. The paper demonstrate that the StableDR estimator has a bounded bias and a bounded generalization bound even when propensities are arbitrarily small. Moreover, the paper proposes a novel cycle learning approach for training a prediction model, a propensity model, and an imputation model based on the StableDR estimator. The key idea of the proposed cycle learning approach is cyclically employing different losses to update the parameters of the three models. The paper conducts extensive experiments on two widely-used datasets containing both Missing Not At Random (MNAT) and Missing At Random (MAR) ratings. The experimental results show that the proposed cycle learning approach based on the StableDR estimator significantly outperforms existing approaches in terms of rating prediction.

**Summary Of The Review:**

I think this paper is marginally below the acceptance threshold due to limited novelty and some not well-supported arguments.

---

> ### Author Response · Authors · 2022-11-17
> **Response to Reviewer f6Th (Part 1 of 5)**
>
> Thank you for the detailed and constructive comments. We addressed all your concerns in our revised manuscript. Specifically, we discussed the difference between the proposed SDR with **DR** and **SNIPS** both **theoretically** and **empirically.** In addition, we added detailed theoretical analysis, especially focusing on the theoretical guarantees for cyclic training orders of the proposed Stable-DR and Stable-MRDR, and conducted extensive real-world experiments to study the effects of varying alternate-update-order empirically in Appendix C. Furthermore, we also added extensive numerical simulations in Appendix D, showing that the proposed SDR has a significantly smaller bias and variance compared to IPS, SNIPS, and DR on eight semi-synthetic datasets with different sample sizes and sparsity. Please kindly find our responses below.
>
> > _**Q1**: The paper argues that existing DR estimators have an unbounded bias, variance, and generalization bound given extremely small propensities, while the proposed StableDR estimator does not. It would be better to formulate this argument formally into some lemma or proposition. To argue that the StableDR estimator is more stable than existing DR estimators, it is crucial to demonstrate that the StableDR estimator has a smaller variance than the DR estimators, both theoretically and empirically._
>
>
> **A1**: Thank you for your helpful suggestions. We provide detailed discussions and demonstrate that SDR estimator has a smaller bias and variance than the DR estimator both **theoretically** and **empirically**.
>
>
> **(a) StableDR estimator has a smaller bias and variance than the DR estimator theoretically**.
>
> On one hand, clearly, if there exist extremely small propensities, the variance of SDR estimator is smaller than the DR estimator, as discussed in Section 3.3.
>   Formally, $\exists ~ C > 0$, if $\hat p_{u, i} < C$ for some $(u, i)\in \mathcal{D}$, then $\text{Bias}(L_{SDR}) < \text{Bias}(L_{DR})$ and $\text{Var}(L_{SDR}) < \text{Var}(L_{DR})$.
>
> On the other hand, we consider the case where there is no extremely small propensities. From [1], it can be shown that
>     $$ \mbox{Var}( L_{DR} )  =  \frac{1}{ |D|^{2} } \sum_{(u,i)\in D}    \frac{p_{u,i} (1- p_{u,i})}{  \hat p_{u,i}^2 } \delta_{u, i}^{2},
>     $$
> where $\delta_{u, i}=\hat e_{u, i}- e_{u, i}$.
> Our Theorem 3 shows that
>     $$
> \mbox{Var}\left(L_{SDR}\right)=\frac{\sum_{(u,i) \in D } p_{u,i}(1-p_{u,i})h^2_{u,i}/\hat p_{u,i}^2}{\left(\sum_{(u,i) \in D}p_{u,i}/\hat p_{u,i}\right)^2}+O( |\mathcal{D}|^{-2} ),
> $$
> where $h_{u,i}=\delta_{u, i}-\sum_{(u,i) \in \mathcal{D}}\{p_{u,i}\delta_{u, i}/\hat p_{u,i}\}/\sum_{(u,i) \in \mathcal{D}}\{p_{u,i}/\hat p_{u,i}\}$.
>
> If $\hat p_{u, i}=p_{u, i}$, or more generally, $\sum_{(u,i) \in \mathcal{D}}p_{u,i}/\hat p_{u,i} \geq \mathcal{D}$, then we have
>     $$
> \mbox{Var}\left(L_{SDR}\right)\leq  \frac{1}{ |D|^{2} } {\sum_{(u,i) \in D} \frac{p_{u,i}(1-p_{u,i})}{\hat p_{u,i}^2} h_{u,i}^2} + O( |\mathcal{D}|^{-2} ),
> $$
> Let $\hat E_\Omega$ be the empirical expectation under the probability measurement $\Omega$, and $$\Omega_1=\\{f_1(u, i)\propto  p_{u,i}(1-p_{u,i})/\hat p^2_{u,i}: (u, i)\in \mathcal{D}\\},\quad \Omega_2=\\{f_2(u, i)\propto  p_{u,i}/\hat p_{u,i}: (u, i)\in \mathcal{D}\\},$$
> where $f_1(u, i)$ and $f_2(u, i)$ are the probability density corresponding to the probability measurement $\Omega_1$ and $\Omega_2$, respectively.
>
> To show $\mbox{Var}\left(L_{SDR}\right) \leq  \mbox{Var}( L_{DR} )$, we now only require that the difference between the empirical expectation of $\delta_{u, i} $ under $\Omega_1$ and $\Omega_2$ is not significant, i.e., $\hat E_{\Omega_2}[\delta_{u, i}]\leq 2\hat E_{\Omega_1}[\delta_{u, i}]$, which is a weak requirement in practice. Then $\mbox{Var}\left(L_{SDR}\right) \leq  \mbox{Var}( L_{DR} )$ follows immediately from the true that
> $
> \hat E_{\Omega_1}[\{\delta_{u, i}- \hat E_{\Omega_2}(\delta_{u, i})\}^2]\leq \hat E_{\Omega_1}[\delta^2_{u, i}].
> $
>
> ***
>
> **References**
>
> [1] Siyuan Guo, Lixin Zou, Yiding Liu, Wenwen Ye, Suqi Cheng, Shuaiqiang Wang, Hechang Chen, Dawei Yin, and Yi Chang. Enhanced doubly robust learning for debiasing post-click conversion rate estimation. In SIGIR, 2021.

---

> > ### Author Response · Authors · 2022-11-17
> > **Response to Reviewer f6Th (Part 2 of 5)**
> >
> > **(b) StableDR estimator has a smaller bias and variance than the DR estimators empirically**.
> >
> > To fully illustrate the performance of the proposed SDR compared to SNIPS and DR, we added extensive Monte Carlo studies in Appendix D, to show that the SDR has smaller bias and variance compared to previous methods empirically. Following the previous studies [1, 2, 3, 4], we simulated four different selection bias levels using **MovieLens 100K (ML-100K)** and **MovieLens 1M (ML-1M)**, respectively, and named them as: **ML-100K-1, ML-100K-2, ML-100K-3, ML-100K-4**, and  **ML-1M-1, ML-1M-2, ML-1M-3, ML-1M-4** with descending order of the propensities scale. Specifically, we set the true propensity $p_{u, i}\in(0,1)$ for each rating $R_{u, i}$, with $p_{u, i}=p \alpha^{1+0.25\cdot \min (4,  5-R_{u, i})\cdot  k}$, where $p=1$, $\alpha=0.5$, $k\in \\{1, 2, 3, 4\\}$, and $R_{u, i} \in\\{1, 2, 3, 4, 5\\}$.
> > The absolute relative error (RE) is used as the evaluation protocol, which is defined as follows
> > $$ \text{RE} (L_{estimator})=  \frac{ | L_{ideal} (\hat{\mathbf{R}}) - L_{estimator} (\hat{\mathbf{R}}) |} {  L_{ideal} (\hat{\mathbf{R}})}, $$
> > where $L_{estimator}$ denotes the estimator to be compared. RE evaluates the accuracy of the estimated loss, the smaller the RE, the more accurate the estimation of the ideal loss. More experimental details are provided in Appendix D.1.
> >
> > We performed 100 repeated experiments on **ML-100K** and  **ML-1M** to compare the estimation performance of Naive, IPS, SNIPS, DR and the proposed SDR under different selection bias levels, and the results are shown in Table 10 in Appendix D. We have the following findings: first, overall, SDR performs significantly better than baselines and the variance is smaller compared to both DR and SNIPS, which empirically demonstrates the stability of our method. Second, there is no significant difference in the performance of IPS and SNIPS in all cases, but our proposed SDR estimator outperforms DR estimator at a statistically significant level of 0.001 in all cases. This suggests that our superior performance is not attributed to self-normalize form, but to carefully designed model dependency, which utilizes the propensity model and imputation model simultaneously in a differentiated way. Third, as the selection bias level increases from **ML-100K/1M-1** to **ML-100K/1M-4**, it can be seen that the estimation error of all baseline estimators increases significantly, while SDR does not. This is attributed to the fact that the proposed SDR estimator is more stable to small propensities.
> >
> > ***
> >
> > **References**
> >
> >
> > [2] Tobias Schnabel, Adith Swaminathan, Ashudeep Singh, Navin Chandak, and Thorsten Joachims. Recommendations as treatments: Debiasing learning and evaluation. ICML, 2016.
> >
> > [3] Xiaojie Wang, Rui Zhang, Yu Sun, and Jianzhong Qi. Doubly robust joint learning for recommendation on data missing not at random. In ICML, 2019.
> >
> > [4] Yuta Saito. Doubly robust estimator for ranking metrics with post-click conversions. In SIGIR, 2020.

---

> > > ### Author Response · Authors · 2022-11-17
> > > **Response to Reviewer f6Th (Part 3 of 5)**
> > >
> > > > _**Q2.1**: I am concerned about the novelty of the proposed approach. First, the StableDR estimator has the same form as an existing Self-Normalized Inverse-Propensity-Scoring (SNIPS) estimator. The property of the StableDR estimator having a bounded bias, variance, and generalization bound mostly comes from the SNIPS estimator._
> > >
> > > **A2.1**: We thank the useful suggestions from the reviewers.
> > >
> > > **(a) Bias-variance comparison on SNIPS and SDR estimators.**
> > >
> > > Theorems 2 and 3 show that
> > > $$
> > > \mbox{Bias}(L_{SDR}) = \Biggl |\frac{1}{|\mathcal{D}|} \sum_{(u,i)\in  \mathcal{D}} \left ( \delta_{u, i}-\frac{\sum_{(u,i)\in  \mathcal{D}}{ \delta_{u, i}p_{u,i}}/\hat p_{u,i}}{\sum_{(u,i)\in \mathcal{D}}{p_{u,i}}/\hat p_{u,i}} \right ) \Biggr | + O( |\mathcal{D}|^{-1} ),
> > > $$
> > > $$
> > > \mbox{Var}\left(L_{SDR}\right)=\frac{\sum_{(u,i) \in \mathcal{D} } p_{u,i}(1-p_{u,i})h^2_{u,i}/\hat p^2_{u,i}}{\left(\sum_{(u,i)\in \mathcal{D}}p_{u,i}/\hat p_{u,i}\right)^2}+O( |\mathcal{D}|^{-2} ),
> > > $$
> > > where $\delta_{u,i} = e_{u,i} - \hat e_{u,i}$, $h_{u,i}=\delta_{u, i}-\sum_{(u,i)\in \mathcal{D}}\{p_{u,i}\delta_{u, i}/\hat p_{u,i}\} \Big / \sum_{(u,i)\in \mathcal{D}}\{p_{u,i}/\hat p_{u,i}\}$.
> > > In addition, when we remove the constraint (3), then SDR estimator reduces to  SNIPS estimator. In Appendix B of the revised version, we show that the bias and variance of the SNIPS estimator and the bias and variance of the SDR estimate have the same form, except that $\delta_{u,i}$ is replaced by $e_{u,i}$. That is,
> > > $$
> > > \mbox{Bias}(L_{SNIPS}) = \Biggl |\frac{1}{|D|} \sum_{(u,i)\in  D} \left ( e_{u, i}-\frac{\sum_{(u,i)\in  D}{ e_{u, i}p_{u,i}}/\hat p_{u,i}}{\sum_{(u,i)\in D}{p_{u,i}}/\hat p_{u,i}} \right ) \Biggr | + O( |D|^{-1} ),
> > > $$
> > > $$
> > > \mbox{Var}\left(L_{SNIPS}\right)=\frac{\sum_{(u,i) \in \mathcal{D}} p_{u,i}(1-p_{u,i}) \tilde h^2_{u,i}/\hat p^2_{u,i}}{\left(\sum_{(u,i)\in \mathcal{D}}p_{u,i}/\hat p_{u,i}\right)^2}+O( |\mathcal{D}|^{-2} ),
> > > $$
> > > where $\tilde h_{u,i}=e_{u,i} -\sum_{(u,i)\in \mathcal{D}}\{p_{u,i}e_{u,i}/\hat p_{u,i}\} \Big / \sum_{(u,i)\in \mathcal{D}}\{p_{u,i}/\hat p_{u,i}\}$.
> > >
> > > Thus, when $|e_{u,i} - \hat e_{u,i} | < e_{u,i}$, that is, $0< \hat e_{u,i} < 2 e_{u,i}$, the bias and variance of SDR estimator tend to smaller than those of SNIPS estimator.
> > >
> > > **(b) SDR has double robustness property, while SNIPS does NOT.**
> > >
> > > It is well known that SNIPS uses a self-normalized form to guarantee robustness to small propensities. However, as shown in Theorem 1, the proposed SDR is robust to small propensities while also having the double robustness property, that is, it is unbiased when either learned propensities or imputed errors are accurate.
> > >  Empirically, doubly robust estimators always outperform "single robust" estimators such as IPS and SNIPS. Thus, combining (a) and (b), the proposed SDR has smaller bias and variance compared to SNIPS when $0< \hat e_{u,i} < 2 e_{u,i}$, as well as double robustness and weaker extrapolation requirements than DR (see the detailed discussion in Section 3.1).
> > >
> > > **(c) Empirical evidence that SDR outperforms SNIPS under varying bias levels.**
> > > We performed a large number of Monte Carlo simulations to demonstrate that the proposed SDR has a significantly better performance compared to SNIPS in all cases. Please refer to response **A1(b)**.

---

> > > > ### Author Response · Authors · 2022-11-17
> > > > **Response to Reviewer f6Th (Part 4 of 5)**
> > > >
> > > > > _**Q2.2**: Second, it is not clear why the paper proposes to train models in a cyclic order of prediction model, imputation model, and propensity model instead of another cyclic order or joint learning of all three models. It is important to add more explanations, justifications, and theoretical or empirical results regarding this aspect._
> > > >
> > > > **A2.2**: We thank the reviewers for their useful comments.
> > > >
> > > > **(a) Motivation for Alternatively Updating the Models.**
> > > >
> > > > From a high-level perspective, learning a recommendation model consists of two closely related tasks:
> > > > - **policy evaluation**:  find an efficient estimator that approximates the ideal loss $L_{ideal}$, and then use the estimation as the evaluation metric of the recommendation model.
> > > > - **policy improvement**: design an efficient algorithm that optimizes the proposed estimator to update the prediction model.
> > > >
> > > >
> > > > For unbiased recommendations learning, previous works (e.g., DR [3] and MRDR [1]) have shown that alternately updating the imputation model (policy evaluation) and prediction model (policy improvement) results in better performance due to their interdependence. Further, recent works (e.g., Learning to Debias [5] and Autodebias [6]) have shown that alternately updating all three models (propensity, imputation, and prediction models) can further improve the debiasing performance. However, our concern is that the latter simply treats the updating of the propensity and imputation model as a whole, without designing the loss separately, which is still essentially joint learning.
> > > >
> > > > In our work, cycle learning is proposed, as shown in Figure 1. We provide more design details, especially focusing on the alternate-update-orders, and demonstrate the advantages of cycle learning over previous work theoretically and empirically as follows (see Appendixes C.2.2 and C.2.3).
> > > >
> > > >
> > > > **(b) Theoretically Guarantees for Alternation Training Orders of the Proposed Stable-DR and Stable-MRDR.**
> > > >
> > > >
> > > > Theoretically, the designed algorithm **strictly follows the proposed SDR estimator** in Section 3.2. As shown in Figure 1 and Alg. 1 in Section 4, our algorithm **first updates imputed errors $\hat e$** by Step 1, and **then learns a propensity $\hat p$** based on learned $\hat e$ to satisfy the constraint (3) in Step 2. The main purpose of the first two steps is to ensure that the SDR estimator in step 3 has double robustness and has a lower extrapolation dependence compared to the previous DR methods. **Finally, from Step 3 we update the predicted rating $\hat r$** by minimizing the estimation of the ideal loss using the proposed SDR estimator. For the next round, instead of re-initializing, Step 1 updates the imputed errors $\hat e$ according to the new prediction model, then Step 2 re-updates the constrained  propensities $\hat p$, and then uses Step 3 to update the prediction model $\hat r$ again. And so on, each model is updated for each Step in our algorithm: Step 1-imputation model $\hat e$, Step 2-propensity model $\hat p$, Step 3-prediction model $\hat r$. Thus the alternate-update-order in our algorithm is acted as $\cdots \longrightarrow \hat e \longrightarrow \hat p \longrightarrow \hat r \longrightarrow \cdots$.
> > > >
> > > >
> > > > **(c) Empirical Performance Comparisons on Varying Alternation Training Orders of the Proposed Stable-DR and Stable-MRDR.**
> > > >
> > > > We conduct extensive experiments to investigate the effect of alternate-update-order on prediction performance using MF, NCF, and SLIM as the base model, Coat and Yahoo as two datasets respectively. Specifically, as in Figure 1, the vanilla DR uses a two-phase learning to update the prediction model without updating alternately. We use DR-JL and MRDR-JL as baselines, which both alternately update $\hat e$ and $\hat r$, while keeping $\hat p$ static. For the proposed Stable-DR and Stable-MRDR, the default alternate-update-order is $\cdots \longrightarrow \hat e \longrightarrow \hat p \longrightarrow \hat r \longrightarrow \cdots$, as stated in Appendix C.2.2. For comparison purposes, we also study Stable-DR and Stable-MRDR performance using an inverse alternate-update-order $\cdots \longrightarrow \hat e \longrightarrow \hat r \longrightarrow \hat p \longrightarrow \cdots$ with $\dagger$ marked.
> > > >
> > > > The empirical results are presented in Tables 4-9. We have the following findings: **First**, same with the previous findings in Appendix C.2.1, updating alternately always leads to a better prediction performance, which is due to performing policy evaluation and policy improvement alternately. **Second**, the proposed Stable-DR and Stable-MRDR significantly outperform DR-JL and MRDR-JL. This is due to the theoretical guarantees of the SDR estimator and our algorithm designs, in particular that the alternate-update-order of the models strictly follows the SDR estimator in Section 3.2 (see the discussion in Appendix C.2.2).
> > > > Compared to DR and MRDR, our method has lower bias and variance in the presence of small propensities.

---

> > > > > ### Author Response · Authors · 2022-11-17
> > > > > **Response to Reviewer f6Th (Part 5 of 5)**
> > > > >
> > > > > **Third**, for the proposed Stable-DR and Stable-MRDR, we find that the regular alternate-update-order $\cdots \longrightarrow \hat e \longrightarrow \hat p \longrightarrow \hat r \longrightarrow \cdots$ has a slightly better performance than using a inverse alternate-update-order $\cdots \longrightarrow \hat e \longrightarrow \hat r \longrightarrow \hat p \longrightarrow \cdots$. This is because the inverse alternate-update-order uses the **outdated information** when updating the prediction model. Specifically, the inverse alternate-update-order is
> > > > > $$ \cdots \longrightarrow \hat e_{t-2} \longrightarrow \hat r_{t-2} \longrightarrow \hat p_{t-2} \longrightarrow \hat e_{t-1} \longrightarrow \hat r_{t-1} \longrightarrow \hat p_{t-1} \longrightarrow \hat e_{t} \longrightarrow \hat r_{t} \longrightarrow \cdots,$$
> > > > > where it should be noted that when updating $\hat r_{t}$, we always use $\hat p_{t-1}$ as the learned propensities instead of $\hat p_{t}$. However, the learning of constrained $\hat p_{t-1}$ uses $\hat e_{t-1}$ as the imputed errors in constraint (3), which is the policy evaluation of $\hat r_{t-2}$. **This results in the update of $r_{t}$ not using the information of $r_{t-1}$ but the information of $r_{t-2}$.** Despite the fact that the inverse alternate-update-order does not perform as well as the regular alternate-update-order, the Stable-DR and Stable-MRDR based on alternate-update-order still stably outperforms the competing DR baselines, because the smaller bias and variance properties are maintained in the presence of small propensities, and the problem for the inverse update is only that it uses outdated information.
> > > > >
> > > > > > _**Q3**: Looks like the property of double robustness in Theorem 1 does not have the exact same definition as that in existing DR estimators. It is recommended to call out the difference and provide some comparisons between these different definition of being double robust._
> > > > >
> > > > > **A3**: Thanks for your comments. In Theorem 1, the double robustness of $\mathcal{E}_{SDR}$ is stated in an **asymptotically unbiased** sense, which is slightly different from existing DR estimators in recommender systems.  ''Asymptotically unbiased'' can be interpreted as ''unbiased" when the sample size goes to infinity.
> > > > >
> > > > > We would like to clarify that the requirements of SDR to achieve unbiasedness has no essential difference comparing to the existing DR methods.  Strictly speaking,  $\hat p_{u,i}$ and $\hat e_{u,i}$ **cannot** be estimated accurately for $p_{u,i}$ and $e_{u,i}$ using fixed sample size, unless the sample size goes to infinity. The existing DR estimators are unbiased under the condition that $\hat p_{u,i}=p_{u,i}$ or $\hat e_{u,i} = e_{u,i}$. However,  the condition "$\hat p_{u,i}=p_{u,i}$" or "$\hat e_{u,i} = e_{u,i}$" is so strong and can only hold in the sense of "asymptotically unbiased", instead of "any given data size".
> > > > >
> > > > > Formally, in statistics, beside the IPS and DR estimators, all "unbiasedness" is expressed in an "asymptotically unbiased" sense  [7, 8, 9, 10].  In recommender systems' community, we usually ignore the notation "asymptotically unbiased" for simplification.
> > > > >
> > > > > ***
> > > > >
> > > > > We would like to thank the useful suggestions from the reviewer. After adding extensive theoretical and experimental analysis in the revised version, we believe we have addressed the reviewers' concerns and welcome any further discussion about our work. We look forward to the reviewer being able to raise the score : )
> > > > >
> > > > > ***
> > > > >
> > > > > **References**
> > > > >
> > > > >
> > > > >
> > > > > [7] Joseph D.Y. Kang and Joseph L. Schafer. Demystifying double robustness: a comparison of alternative strategies for estimating a population mean from incomplete data. Statistical Science, 22:523–539, 2007.
> > > > >
> > > > > [8] Zhiqiang Tan. Comment: understanding OR, PS and DR. Statistical Science, 22:560–568, 2007.
> > > > >
> > > > > [9]  Vermeulen, K. and Vansteelandt, S. (2015). Bias-reduced doubly robust estimation. Journal of the American Statistical Association 110, 1024–1036.
> > > > >
> > > > > [10]  Shaun R. Seaman and Stijn Vansteelandt (2018). Introduction to Double Robust Methods for Incomplete Data. Statistical Science, 33:184-197, 2018.

---

> ### Author Response · Authors · 2022-11-21
> **Thanks for your constructive comments and we would like to address your concerns and questions.**
>
> We thank the reviewer for the detailed and constructive comments. Specifically, we have added extensive theoretical analysis and experiments, with special focus on the bias-variance theoretical analyses compared to standard SNIPS and DR, as well as the order of cyclic update of the models. We hope that these revisions could address your concerns and we appreciate if you could get back to us. Thank you!

---

### Official Review · Reviewer_si5q · 2022-10-25

**Confidence:** 3
**Correctness:** 4
**Technical Novelty And Significance:** 3
**Empirical Novelty And Significance:** 3
**Recommendation:** 8

**Clarity, Quality, Novelty And Reproducibility:**

The paper is clearly written, the theoretical and experiment results are strong and there is no concern about reproducibility given the code in the supplementary file.
Theorem 1 appears to be novel and the remainder of the paper follows from that simple insight.


**Strength And Weaknesses:**

Strengths:

1. The proposed approach in Algorithm 1 is simple, well-motivated and backed by strong theoretical results.

2. The experimental section is strong. For instance, ablation studies on the stabilization constraint in Sec 5.3 demonstrate its efficacy despite the use of two different propensity estimators.

Weaknesses:

1. It is not clear why NCF is used as a baseline in Table 1. In "Are we really making much progress? A worrying analysis of recent neural recommendation approaches", RecSys 2019, it was shown that the SLIM baseline outperforms NCF.

2. The authors ignore temporal variability in recommender systems and assume that features $x_{ui} $ for a user-item pair are static rather than temporally varying.
Recurrent Recommender Networks, WSDM 2017, shows that this assumption often does not hold for several real world recommender systems.

3. Typos exist, e.g., in the sentence before Theorem 3, "when there exist an..." should be "when there exists an ... "


**Summary Of The Paper:**

By applying a stabilizing constraint on propensity model learning given an error imputation model, the authors show that the result SNIPS estimator satisfies the double robustness property in Theorem 1. Hence this estimator is termed the SDR estimator. This naturally results in bounded bias (Theorem 2), a tail bound on the deviation (Theorem 4) and a generalization bound (Theorem 5), demonstrating mitigation of the effects caused by small propensities.
Motivated by these theoretical results, for recommender systems, the authors propose iterative 3-step alternating optimization of the error imputation model, the propensity model and the prediction model. SDR and SMRDR outperform competitors on two public rating datasets using MF and NCF baselines for the prediction model.



**Summary Of The Review:**

Despite the concern regarding general practical applicability mentioned under weaknesses, given the simple elegance of the proposed approach, the strong theoretical foundations and the convincing results, I recommend acceptance of this paper.

---

> ### Author Response · Authors · 2022-11-17
> **Response to Reviewer si5q (Part 1 of 3)**
>
> Thank you for the positive and constructive comments. We addressed all your concerns and added extensive real-world experiments using **SLIM** as the base model (see Table 1 in the revised version), and studied the alternate-update-order effect of models on the debiasing performance using both **Coat** and **Yahoo** (see Appendix C in the revised version). Please kindly find our responses below.
>
>
> > _**Q1**: It is not clear why NCF is used as a baseline in Table 1. In "Are we really making much progress? A worrying analysis of recent neural recommendation approaches", RecSys 2019, it was shown that the SLIM baseline outperforms NCF._
>
> **A1**: We fully agree with the reviewer and remove the NCF in the main text to Appendix C.1, using SLIM as a replacement. Surprisingly, from Table 1 in the revised version, SLIM has significantly higher NDCG@5 and NDCG@10 compared to MF and NCF. In addition, SLIM has higher parallelism, which would empirically result in a faster runtime. After comparing with a wide range of competitive baselines, we find that the proposed Stable-DR and Stable-MRDR still stably outperform the DR and MRDR methods on both of the datasets.
>
> Besides, to fully illustrate the performance of the proposed SDR compared to SNIPS and DR, we added extensive Monte Carlo studies in Appendix D, to show that the SDR has smaller bias and variance compared to previous methods empirically. Following the previous studies [1, 2, 3, 4], we simulated four different selection bias levels using **MovieLens 100K (ML-100K)** and **MovieLens 1M (ML-1M)**, respectively, and named them as: **ML-100K-1, ML-100K-2, ML-100K-3, ML-100K-4**, and **ML-1M-1, ML-1M-2, ML-1M-3, ML-1M-4** with descending order of the propensities scale. Specifically, we set the true propensity $p_{u, i}\in(0,1)$ for each rating $R_{u, i}$, with $p_{u, i}=p \alpha^{1+0.25 \cdot \min (4,  5-R_{u, i})\cdot k}$, where $p=1$, $\alpha=0.5$, $k\in \\{1, 2, 3, 4\\}$, and $R_{u, i} \in\\{1, 2, 3, 4, 5\\}$.
> The absolute relative error (RE) is used as the evaluation protocol, which is defined as follows
> $$ \text{RE} (L_{estimator})=  \frac{ | L_{ideal} (\hat{\mathbf{R}}) - L_{estimator} (\hat{\mathbf{R}}) |} {  L_{ideal} (\hat{\mathbf{R}})},$$ where $\mathcal{E}_{{estimator}}$ denotes the estimator to be compared. RE evaluates the accuracy of the estimated loss, the smaller the RE, the more accurate the estimation of the ideal loss. More experimental details are provided in Appendix D.1.
>
> We performed 100 repeated experiments on **ML-100K** and  **ML-1M** to compare the estimation performance of Naive, IPS, SNIPS, DR and the proposed SDR under different selection bias levels, and the results are shown in Table 10 in Appendix D. We have the following findings: first, overall, SDR performs significantly better than baselines and the variance is smaller compared to both DR and SNIPS, which empirically demonstrates the stability of our method. Second, there is no significant difference in the performance of IPS and SNIPS in all cases, but our proposed SDR estimator outperforms DR estimator at a statistically significant level of 0.001 in all cases. This suggests that our superior performance is not attributed to self-normalize form, but to carefully designed model dependency, which utilizes the propensity model and imputation model simultaneously in a differentiated way. Third, as the selection bias level increases from **ML-100K/1M-1** to **ML-100K/1M-4**, it can be seen that the estimation error of all baseline estimators increases significantly, while SDR does not. This is attributed to the fact that the proposed SDR estimator is more stable to small propensities.
>
> **References**
>
> [1] Tobias Schnabel, Adith Swaminathan, Ashudeep Singh, Navin Chandak, and Thorsten Joachims. Recommendations as treatments: Debiasing learning and evaluation. ICML, 2016
>
> [2] Xiaojie Wang, Rui Zhang, Yu Sun, and Jianzhong Qi. Doubly robust joint learning for recommendation on data missing not at random. In ICML, 2019.
>
> [3] Yuta Saito. Doubly robust estimator for ranking metrics with post-click conversions. In SIGIR, 2020
>
> [4] Siyuan Guo, Lixin Zou, Yiding Liu, Wenwen Ye, Suqi Cheng, Shuaiqiang Wang, Hechang Chen, Dawei Yin, and Yi Chang. Enhanced doubly robust learning for debiasing post-click conversion rate
> estimation. In SIGIR, 2021

---

> > ### Author Response · Authors · 2022-11-17
> > **Response to Reviewer si5q (Part 2 of 3)**
> >
> > > _**Q2**: The authors ignore temporal variability in recommender systems and assume that features_  $x_{u,i}$ _for a user-item pair are static rather than temporally varying. Recurrent Recommender Networks, WSDM 2017, shows that this assumption often does not hold for several real-world recommender systems._
> >
> > **A2**: We thank the reviewer for the inspiring and constructive comments. Following previous studies [1, 2, 3, 4], in this paper, we only focus on the study of static user-item pairs and embeddings. To the best of our knowledge, there are no open source datasets containing both of the timestamps and missing-at-random (MAR) ratings, therefore we use two commonly used open source datasets **Coat** and **Yahoo** containing missing-not-at-random (MNAR) and MAR ratings for the evaluation protocol in our real-world experiments. **However, it should be noted that the proposed methods can be naturally extended to temporal selection bias settings both theoretically and empirically.**
> >
> > **Theoretically,** for the setting of temporal debiased recommendation, we consider time-varying users' true preferences $r_{u, i, t}(1)$, where $u \in \mathcal{U}$ and $i \in \mathcal{I}$ are the sets of indexes for users and items, respectively, and $t \in \mathcal{T}$ is the timestamp. Then for a given prediction model $\hat r_{u, i, t}=f(x_{u, i, t};\phi)$, the ideal loss is
> > \begin{equation}
> >     L_{ideal}(\phi)  =  \frac{1}{|\mathcal{U}|\cdot|\mathcal{I}|\cdot|\mathcal{T}|}  \sum_{(u,i,t) \in \mathcal{U}\times  \mathcal{I}\times  \mathcal{T}} e_{u,i,t},  %= \bfE({e_{u,i}}),
> >   \end{equation}
> > where $e_{u,i,t}$ is the temporal prediction error, such as the squared loss  $e_{u,i, t} = (\hat r_{u,i, t}(1)  - r_{u,i, t}(1))^2$.  Similarly, $L_{ideal}(\phi)$ can be regarded as a benchmark of unbiased temporal loss function, even though it is infeasible due to the missingness of $\\{ r_{u,i, t}(1): o_{u,i, t} = 0\\}$, where $o_{u, i, t}=1$ if user $u$'s rating of item $i$ at time $t$ has been recorded in the logged data, and $o_{u, i, t}=0$ otherwise. Then the proposed stabilized doubly robust (SDR) estimator can be naturally extended to the above temporal selection bias setting, containing the following three steps.
> >
> > **Step 1 (Initialize **temporal** imputed errors).** Pre-train a temporal imputation model $\hat e_{u,i,t}$ and let $\mathcal{\hat E} \triangleq \sum_{(u,i,t) \in \mathcal{U}\times  \mathcal{I}\times  \mathcal{T}} \hat e_{u,i,t}$.
> >
> > **Step 2 (Learn **temporal** constrained propensities.** Learn a temporal propensity model $\hat p_{u,i, t}$ satisfying
> > $$
> > \frac{1}{|U|\cdot|I|\cdot|T|} \sum_{(u,i,t) \in U \times I \times  T} {\frac{o_{u,i, t}}{\hat p_{u,i, t}}}{\left(\hat e_{u,i, t}- \mathcal{\hat E}\right)}   = 0.
> > $$
> >
> >
> > **Step 3 (**Temporal** SDR estimator).** The temporal SDR (T-SDR) estimator is given as
> > $$
> > L_{T-SDR}
> > = \sum_{(u,i,t) \in U\times  I \times  T} \frac{o_{u,i,t} e_{u,i,t}}{\hat p_{u,i,t}} \\Big  / \sum_{(u,i,t) \in U\times I \times T} \frac{o_{u,i,t}}{\hat p_{u,i,t}}.
> > $$
> >
> > It is straightforward that the T-SDR estimator has double robustness and has small bias, variance, and generalization error bounds in the presence of small propensities.

---

> > > ### Author Response · Authors · 2022-11-17
> > > **Response to Reviewer si5q (Part 3 of 3)**
> > >
> > > **Empirically,** for the temporal propensity model training in Step 2, the backbone can be a temporal matrix decomposition can be used as the skeleton, e.g., Time-aware matrix factorization (TMF) [5], i.e., $\hat p_{u, i, t}=\sigma (p_u^T q_i+a_t)$,
> > >  or Time-aware tensor factorization (TTF) [6], i.e., $\hat p_{u, i, t}=\sigma(p_u^T (q_i \times b_t))$,
> > >  where $p_u$ and $q_i$ are the embeddings of user $u$ and item $i$, respectively, and $a_t$ and $b_t$ account for the user's time-varying interests. The learned temporal propensities need to both with high accuracy, which is evaluated with cross entropy, and meet the constraint in Step 2 for stabilization and double robustness. Therefore the temporal propensity model is updated by minimizing
> > > $$
> > > L_p(\hat p_{u, i, t})=\sum_{(u,i,t) \in U \times I \times T} o_{u, i, t} \cdot \log \hat p_{u, i, t}+(1-o_{u, i, t}) \cdot \log (1-\hat p_{u, i, t})+ \eta \cdot \left \\{\sum_{(u,i,t) \in U \times  I \times T} \frac{o_{u,i,t}}{\hat p_{u, i, t}}{\left(\hat e_{u,i,t}-\mathcal{\hat E}\right)} \right \\}^2,
> > > $$
> > > where $\eta$ is a hyper-parameter for the trade-off between accuracy and constraint in the Step 2.
> > >
> > > For the temporal error imputation model in Step 1 and the temporal prediction model in Step 3, the backbone can be a  Recurrent Recommender Networks (RRN, in WSDM 2017), as suggested by the reviewer. Specifically, the temporal error imputation model is updated by minimizing
> > > $$
> > > L_{e}(\hat e_{u, i, t})=\frac{1}{|U|\cdot|I|\cdot|T|} \sum_{(u,i,t) \in U\times  I\times  T} \frac{ o_{u,i,t} ( \hat e_{u, i,t}-e_{u, i,t})^{2}  }{\hat p_{u,i,t}} $$
> > > and the temporal prediction model is updated by minimizing
> > > $$
> > > L_r(\hat r_{u, i, t})=\sum_{(u,i,t) \in U\times  I\times  T} \frac{o_{u,i,t} e_{u,i,t}}{\hat p_{u,i,t}}  / \sum_{(u,i,t) \in U\times  I\times  T} \frac{o_{u,i,t}}{\hat p_{u,i,t}}.
> > > $$
> > > We leave A/B testing that considers temporal selection bias for future work.
> > >
> > >
> > > > _**Q3**: Typos exist, e.g., in the sentence before Theorem 3, "when there exists an..." should be "when there exists an ... ": The definition of being double robust._
> > >
> > >
> > > **A3**: Thanks for pointing this out. It has been changed.
> > >
> > > ***
> > >
> > > We hope that we have addressed the reviewer's concerns, and look forward to future engagement about our work.
> > >
> > > ***
> > >
> > > **References**
> > >
> > > [5] Yehuda Koren. Collaborative Filtering with Temporal Dynamics. In KDD, 2009.
> > >
> > > [6] Liang Xiong, Xi Chen, Tzu-Kuo Huang, Jeff Schneider, and Jaime G Carbonell. Temporal Collaborative Filtering with Bayesian Probabilistic Tensor Factorization. In SDM. SIAM, 2010.

---

> > > > ### Comment · Reviewer_si5q · 2022-11-21
> > > > **Response**
> > > >
> > > > I wish to thank the authors for their thoughtful and detailed response as well as the additional experimental results on SLIM and theoretical extensions regarding temporal variability. (My high score remains unchanged.)

---

### Author Response · Authors · 2022-11-17
**Thanks to the Reviewers and Revised Manuscript Submitted.**

We thank the reviewers for their insightful feedback and comments. We have added extensive theoretical analysis and experiments as suggested by the reviewers and have made appropriate changes to our manuscript. We provide a brief list of the manuscript revisions and additional experiments below.
- 1. We added real-world experiments using SLIM as prediction backbone (see Table 1 in the revised version) as requested by reviewer si5q, and removed the NCF experimental results to Appendix C.1.
- 2. As requested by reviewer f6Th, we added detailed theoretical analysis, especially focusing on the theoretical guarantees for cyclic training orders of the proposed Stable-DR and Stable-MRDR, and conducted extensive real-world experiments to study the effects of varying alternate-update-order empirically in Appendix C.2.
- 3. As requested by the reviewer f6Th and HkXH, we also added extensive numerical simulations in Appendix D, showing that the proposed SDR has a significantly smaller bias and variance compared to IPS, SNIPS, and DR on eight semi-synthetic datasets with different sample sizes and sparsity.
- 4. We performed a comprehensive theoretical analysis to compare the proposed SDR with previous SNIPS and DR, please refers to the detailed response to the reviewers.

We hope that our response addresses the reviewers' concerns and look forward to further discussion.

---

### Decision · Program_Chairs · 2023-01-20

**Decision:**

Accept: poster

**Justification For Why Not Higher Score:**

Reviewers share a concern that SDR estimator is similar to the existing SNIPS estimator. While the authors provided new theoretical analysis showing the difference (which is important and convincing), it requires substaintial revision of the paper.

**Justification For Why Not Lower Score:**

The proposed SDR estimator is novel and well-motivated. Theoretical results clearly showed the advantage of the proposed SDR estimator.  All reviewers agree that the experiments are thorough and convincing. Given the high rating, I am pleased to recommend acceptance.

**Metareview: Summary, Strengths And Weaknesses:**

The authors propose stabilized doubly robust (SDR)  method for learning a recommender system from Missing Not At Random (MNAR) feedback data. The authors showed that the SDR estimator has a bounded bias and a bounded generalization bound even when propensities are arbitrarily small. The authors further proposed cycle learning that alternatively optimizing the error imputation model, the propensity model and the prediction model. Experimental results showed that StableDR and StableMRDR outperform competitors on two public rating datasets using MF and NCF baselines for the prediction model. The authors provided extensive theoretical analysis and experiments during the discussion period. Theoretical results in reivsed version clearly showed the advantage of the proposed SDR estimator,  All reviewers agree that the experiments are thorough and convincing. I am pleased to recommend acceptance. The authors are encouraged to include the discussion with reviewers in the final version, such as clarifying the difference between SDR and SNIPS.



**Note From Pc:**

if the above contains the word "oral" or "spotlight" please see: "oral" presentation means -> notable-top-5% and "spotlight" means -> notable-top-25%. As stated in our emails, we are disassociating presentation type from AC recommendations